# Barnacles Mating Optimizer Algorithm to Extract the Parameters of the Photovoltaic Cells and Panels

**DOI:** 10.3390/s22186989

**Published:** 2022-09-15

**Authors:** Manoharan Madhiarasan, Daniel T. Cotfas, Petru A. Cotfas

**Affiliations:** Department of Electronics and Computers, Faculty of Electrical Engineering and Computer Science, Transilvania University of Brasov, 500036 Brasov, Romania

**Keywords:** photovoltaic cells and panels, barnacles mating optimizer algorithm, statistical tests, parameters, extract

## Abstract

The goal of this research is to accurately extract the parameters of the photovoltaic cells and panels and to reduce the extracting time. To this purpose, the barnacles mating optimizer algorithm is proposed for the first time to extract the parameters. To prove that the algorithm succeeds in terms of accuracy and quickness, it is applied to the following photovoltaic cells: monocrystalline silicon, amorphous silicon, RTC France, and the PWP201, Sharp ND-R250A5, and Kyocera KC200GT photovoltaic panels. The mathematical models used are single and double diodes. Datasets for these photovoltaic cells and panels were used, and the results obtained for the parameters were compared with the ones obtained using other published methods and algorithms. Six statistical tests were used to analyze the performance of the barnacles mating optimizer algorithm: the root mean square error mean, absolute percentage error, mean square error, mean absolute error, mean bias error, and mean relative error. The results of the statistical tests show that the barnacles mating optimizer algorithm outperforms several algorithms. The tests about the computational time were made using two computer configurations. Using the barnacles mating optimizer algorithm, the computational time decreases more than 30 times in comparison with one of the best algorithms, hybrid successive discretization algorithm.

## 1. Introduction

The increases in the demand for electric and thermal energy, the political issues, and the supply, especially with natural gas, have led to an accelerated price growth in recent times. Additionally, the fight against climate change also creates a new perspective for renewable energy, especially for the electric energy produced by photovoltaic panels.

According to the report (the newest) presented in [1], from 2017 to 2020, the PV installed capacity exceeded the wind installed capacity, proving that the conversion of solar energy into electric energy using photovoltaic panels is now the leading actor in renewable energies.

Knowing the photovoltaic cells and panel parameters is vital for the researchers, manufacturers, and photovoltaic systems designers [2]. The current–voltage characteristic, I-V, the equivalent circuit, the mathematical models, analytical methods, and metaheuristic algorithms are used to extract the parameters of the photovoltaic cells and panels (PV) [3]. The complexity of the equivalent circuit is provided by the considered mechanisms for the photovoltaic cells and, in consequence, the number of the diode in equivalent circuit. The most used model is the single diode model (SDM) due to its simplicity, and reduced number of parameters, as only five must be extracted. Due to the increasing performance of the technologies, the two-diode model (DDM) has become more used lately. Even though the extraction process of the photovoltaic cells and panels parameters began at the beginning of the 1960s [4], it is still very popular because of the metaheuristic algorithms and artificial intelligence [3]. The first advantage of the metaheuristic algorithms is that they allow extracting all five parameters for the SDM model or seven parameters for the DDM model, while a lot but not most analytical methods allow determining one or more, but not all parameters.

Some papers reviewed the method used to extract the parameters of the photovoltaic cells and panels. Cotfas et al. discussed 35 methods to extract the PV parameters, pros, and cons. Most of them are analytical, but two metaheuristic algorithms, particle swarm optimization and genetic algorithms, are discussed [4]. Humada et al. reviewed the analytical methods function of the number of extracted parameters for both models SDM and DDM [5]. Yang et al. achieved a comprehensive review to analyze twenty-eight metaheuristic algorithms. They classified the metaheuristic algorithms into four groups: mathematics algorithms, sociology algorithms, physics algorithms, and biology algorithms [3]. An analysis of the methods to extract the parameters of 17 photovoltaic cells and panels was achieved by Pillai et al. Following this analysis, the best method/algorithm was found for each PV device and their use to detect the fault [6]. Li et al. reviewed the metaheuristic algorithms used to identify the parameters of PV devices analyzing the reliability, computational resources, and complexity of each algorithm [7]. An improvement of the single diode model was presented by Rawa et al. Additionally, the results RMSE for RTC France photovoltaic cell using various algorithms were shown for the three models SDM, DDM, and TDM [8].

By analyzing the review papers and the specialized literature, several algorithms were selected to compare the results with the ones provided by the barnacles mating optimizer algorithm (BMOA) considering the accuracy and the extracting time, which implicitly means the number of iterations necessary for the algorithm to extract the parameters.

The shuffled complex evolution (SCE) algorithm is part of the mathematical group. It uses four concepts: the combination of random and deterministic approaches, clustering, systematic evolution, and competitive complex evolution CCE, which is the critical component of SCE [9]. Pattern search (PS) is an algorithm without derivatives, independent of the initial point, without approximations for the objective function, and it uses its search history for new search ways to optimize the process [10]. Using the discretization technique, the successive discretization (SDA) algorithm very accurately extracts the parameters of the photovoltaic cell from the current–voltage pairs measured, but the extraction time is high [11]. The extracting time was reduced using the parallelization successive discretization (PSDA) algorithm [12], and for several photovoltaic cells and panels the accuracy was improved using hybridization algorithm HSDA which uses two algorithms, one from specialized literature and the second being SDA [13].

Imperialist competitive algorithm (ICA) [14] is part of the sociology algorithms group. It is based on the conquest process by the powerful empires of the less powerful. The convergence speed is high because the number of iterations is low 100, especially for low dimensional optimizations, but the accuracy is relatively low.

The physics group contains some algorithms, such as the simulated annealing algorithm (SA) [15], where the crystallization process is adapted so the objective function is minimized, the accuracy is low, and the extracting time is high; particle swarm optimization (PSO) based on the foraging behavior of birds which is improved for increasing its accuracy using hybridization [16], chaos optimization algorithm (COA) which has a short execution time [17], and wind driven optimization (WDO) based on study of the velocity and position of microscopic air parcels in space, its accuracy and convergence time is similar with SA, GA, and PS [18]. The last three algorithms were an important impact due to their flexibility to be improved and enter hybridization with other algorithms. Several new algorithms which are based on PSO, COA, and WDO are: chaotic particle swarm optimization algorithm (CPSO) [19], guaranteed convergence particle swarm optimization (GCPSO) [20], time-varying acceleration coefficients particle swarm optimization (TVACPSO) [21], mutative scale parallel chaos optimization algorithm (MPCOA) [22], and WDOWOAPSO which combines the abilities of the WDO and PSO [23].

The biology group contains a lot of algorithms, both simple and hybrid. The first algorithm was the genetic algorithm (GA) which is based on selection, crossover, and mutation [24,25,26]. Jervase et al. used for the first time the genetic algorithm to extract the photovoltaic cell parameters and the results obtained were better than most of the methods, but with modest results in comparison with those obtained using the latest metaheuristic algorithms [24]. Bastidas-Rodriguez et al. used the GA to extract parameters of only one photovoltaic panel BP585, which is mono-crystalline, used for the population size of 20 [25]. Cotfas et al. used GA to extract the parameters using SDM for two photovoltaic cells and one panel. It was implemented in the R programming language. The population size used was 500, and the iterations number was 5000 [26]. For comparison, other algorithms are used: the flower pollination algorithm (FPA) based on the flower pollination process with low to medium accuracy [27]; the grey wolf optimization (GWO) algorithm, which uses the methods of grey wolves when hunting with a strong ability to find the global optimum [28]; the exploration-enhanced grey wolf optimization (EEGWO) which has good results, but uses a high number of iterations, 50,000, and consequently a high parameters extracting time [29]; and cuckoo search (CS) which is used to extract parameters of the photovoltaic cells using only the SDM model [30].

The analytical five-point method (5P) is used for comparison due to its good performance, being one of the best analytical methods for photovoltaic cell parameters extraction [31].

The novelty and contributions of this paper are:Implementing a new metaheuristic algorithm for the first time to extract the five or seven parameters of three photovoltaic cells and for three photovoltaic panels, barnacles mating optimizer algorithm (BMOA);Comparison of the results obtained using BMOA with the algorithms chosen using the statistical tests and the number of iterations;Comparison of the results when the number of the epochs and the population are changed;Analyzing the computational time necessary to extract the parameters of photovoltaic cells and panels function of the computer configuration;BMOA is an algorithm that provides excellent results for the statistical test applied compared with the algorithms from the specialized literature;The extracting time is reduced more than 30 times when the barnacles mating optimizer algorithm (BMOA) is used in comparison with SDA algorithm, so the BMOA algorithm comes to fill the gap for the algorithms with high accuracy and a short time to extract the parameters of the photovoltaic cells and panels.

The paper is structured in four sections: introduction, method and algorithm description, results and discussions, and conclusion.

## 2. Method and Algorithm Description

### 2.1. PV Models

The SDM and DDM models are used in this paper to extract the parameters of the monocrystalline silicon, amorphous silicon, RTC France photovoltaic cells, Sharp ND-R250A5, PWP201, and Kyocera KC200GT photovoltaic panel.

The single diode model (SDM) allows for the determination of five parameters using Equation (1) the photogenerated current *I_ph_*, the reverse saturation current *I*_0_, the series resistance *R_s_*, the shunt resistance *R_sh_*, and the ideality factor of diode *n*.
(1)$$I=I_{ph}-I_{o}\left(e^{\frac{V+IR_{s}}{nV_{T}}}-1\right)-\frac{V+IR_{s}}{R_{sh}}$$
where *V_T_* = *kT*/*q* is thermal voltage, *q* = 1.60217646 × 10^−19^ is electric elementary charge, *k* = 1.3806503 × 10^−23^ is Boltzmann constant, and *T* is temperature of the photovoltaic device.

The SDM model is used to extract the parameter when necessary, fast response, and a low manufacturing cost [3].

The two-diode model considers the diffusion and generation-recombination mechanism which takes place in the photovoltaic cells and panels. The DDM model provides very good results, and it must be used for low irradiation levels and to have a high curve fitting accuracy [3]. The mathematical model for the photovoltaic panels is presented in Equation (2).
(2)$$I=N_{p}I_{ph}-N_{p}I_{01}\left(e^{\frac{N_{p}V+N_{s}IR_{s}}{n_{1}N_{p}N_{s}V_{T}}}-1\right)-N_{p}I_{02}\left(e^{\frac{N_{p}V+N_{s}IR_{s}}{n_{2}N_{p}N_{s}V_{T}}}-1\right)-\frac{N_{p}V+N_{s}IR_{s}}{N_{s}R_{sh}}$$
where *N_p_* represents the number of the photovoltaic cells connected in parallel, *N_s_* represents the number of the photovoltaic cells connected in series, index (1) is for the diffusion mechanism, and (2) is for the generation-recombination mechanism. If *N_p_* and *N_s_* are equal 1, Equation (2) becomes DDM model for photovoltaic cells.

### 2.2. Barnacles Mating Optimizer Algorithm

Many metaheuristic algorithms are used to extract the parameters of the photovoltaic cells and panels to increase the accuracy of their analysis and characterization.

Sulaiman et al. analyzing the barnacles’ mating action, developed the Barnacles Mating Optimizer (BMO) Algorithm. It is a metaheuristic optimization algorithm [32].

#### 2.2.1. Barnacles Mating Optimizer Pseudo

Step 1: Initialize the optimization algorithm with respect to the population size, length of the penis, upper and lower bound, and so on;

Step 2: Perform the objective function calculation for each and every barnacle;

Step 3: Perform sorting in descending order from the largest length of the penis to the shortest; the optimal result is position on the top population;

Step 4: Perform the iteration process

While (epoch < Max epoch),

Based on the dad and mom selection equation, choose the dad and mom of barnacles.

If explored, DadBarnacles and MomBarnacles = length of penis set value (LP).

For each variable,

Generate off-spring

end for

else if explored DadBarnacles and MomBarnacles > LP

for each variable,

generate off-spring Equation (8)

end for

end if

if the solution is out of boundaries, obtain the current barnacles

compute objective function

perform sorting and updating process

e = e + 1

end while

return as the optimal solution.

Step 5: Record the result and end.

#### 2.2.2. Mathematical Modeling of Barnacles Mating Optimizer Algorithm

Candidate solution (3):(3)$$B=\left[\begin{array}{ccccccccc} B_{1}^{1} & . & . & . & . & . & . & B_{1}^{C} & \\ . & . & . & . & . & . & . & . & \\ . & . & . & . & . & . & . & . & \\ . & . & . & . & . & . & . & . & \\ . & . & . & . & . & . & . & . & \\ . & . & . & . & . & . & . & . & \\ . & . & . & . & . & . & . & . & \\ B_{t}^{1} & . & . & . & . & . & . & . & B_{t}^{C} \end{array}\right]$$ where *C* is the control variables numbers, and *t* is the number of barnacles.

Upper and lower bounds ((4) and (5)):(4)$$ub=\left[ub_{1}\;.\;.\;.\;ub_{n}\right]$$
(5)$$lb=\left[lb_{1}\;.\;.\;.\;lb_{n}\right]$$
where *n* is the variable.

Selection of Dad and Mom Barnacles:(6)$$Dad_{Barnacles}=rand_{-}sperm\left(t\right)$$
(7)$$Mom_{Barnacles}=rand_{-}sperm\left(t\right)$$

Barnacles off-spring generation:(8)$$B_{n}^{C,new}=xB_{Dad_{Barnacles}}+yB_{Mom_{Barnacles}}$$
where *x* is the normal distributed pseudo random number (0, 1) and *y* is provided by Equation (9):(9)$$y=\left(1-x\right)$$
where *LP* is the Length of the Penis

Process of sperm cast:(10)$$B_{n}^{t\_new}=rand\left(\right)*B_{Mom_{Barnacles}}^{t}$$
where *rand*( ) is the random number between 0 and 1. The proposed BMO algorithm-based PV cell/panel parameters extraction pictorial overview is illustrated in Figure 1.

The proposed BMOA set parameter values are tabulated in Table 1.

### 2.3. Objective Function and Statistical Tests

The goal is to extract the PV parameter (five or seven) with better accuracy and minimize the errors between the measured data and calculated ones. The objective function is defined as the minimization of the RMSE and the mathematical equation presented in Equation (11).
(11)$$RMSE_{\left(I,V\right)}=\sqrt{\frac{\sum_{i=1}^{p}{\left(I_{ic}-I_{im}\right)}^{2}}{p}}$$
where *p* represents the number of the measurement points, *I_ic_* is the calculated current, and *I_im_* is the measured current.

The statistical errors are computed between calculated and measured current using the Equations (12)–(16). The following statistical metrics are used: mean absolute percentage error (MAPE), mean square error (MSE), mean absolute error (MAE), mean bias error (MBE), and mean relative error (MRE).
(12)$$MAPE_{\left(I,V\right)}=\frac{100\sum_{i=1}^{p}\left|\frac{\left(I_{ic}-I_{im}\right)}{{\bar{I}}_{im}}\right|}{p}$$
(13)$$MSE_{\left(I,V\right)}=\frac{\sum_{i=1}^{p}{\left(I_{ic}-I_{im}\right)}^{2}}{p}\;$$
(14)$$MAE_{\left(I,V\right)}=\frac{\sum_{i=1}^{p}\left(\left|I_{ic}-I_{im}\right|\right)}{p}\;$$
(15)$$MBE_{\left(I,V\right)}=\frac{\sum_{i=1}^{p}\left(I_{ic}-I_{im}\right)}{p}\;$$
(16)$$MRE_{\left(I,V\right)}=\frac{\sum_{i=1}^{p}\left|\frac{\left(I_{ic}-I_{im}\right)}{{\bar{I}}_{im}}\right|}{p}\;$$
where ${\bar{I}}_{im}$ is the average of the measured current.

## 3. Results and Discussions

Six datasets from the specialized literature are used to study the performance of the BMOA algorithm in accuracy, the number of iterations, and computational time for three commercial photovoltaic cells: monocrystalline silicon, amorphous silicon, and RTC France, and three photovoltaic panels: PWP201, Sharp ND-R250A5, and Kyocera KC200GT.

### 3.1. Monocrystalline Silicon Photovoltaic Cell

Monocrystalline silicon photovoltaic cell is a commercial one sized 3 cm × 3 cm, and the current–voltage characteristic was measured at 27 °C under illumination at 1000 W/m^2^.

The (*I*, *V*) pairs, the calculated current using SMD and DDM models, and the errors are presented in Table 2.

Using BMOA algorithm, the parameters of the monocrystalline silicon photovoltaic cell are extracted for both models considering a range for each parameter, see Table 3.

By analysis, the statistical test for the algorithms, especially RMSE, the BMOA surpasses the genetic algorithm and the five parameters methods, but SDA for SDM and PSDA for DDM have better results, as seen in Table 4. The MAPE and MRE present better results for the BMOA algorithm in comparison with all other algorithms.

The extracted parameters are presented in Appendix A. The highest difference is for the shunt resistance and for the ideality factor of the diode. For example, for shunt resistance, the difference is 100% in comparison with SDA.

The current–voltage characteristics measured and calculated are presented in Figure 2a for the SDM model and Figure 2b for the DDM model. The errors calculated for the calculated current are presented in Figure 2c for both models.

The matching between the two current–voltage characteristics is very good for each model, but upon analyzing the errors it can be observed that there is a higher error around the short circuit current, maximum power, and open circuit voltage. The SDM presents lower values of the errors around the short circuit current and open circuit voltage than DDM, but the behavior is changed drastically around the maximum power.

### 3.2. Amorphous Silicon Photovoltaic Cell

The current–voltage characteristic of the amorphous silicon photovoltaic cell, aSi, was measured at 25 °C temperature. Keitley 4200 is used to measure I-V characteristic. The level of irradiance was 1000 W/m^2^ obtained using LOT Oriel solar simulator.

The (I, V) pairs, the calculated current using SMD and DDM models, and the errors are presented in Table 5.

The parameters of the amorphous silicon photovoltaic cell extracted for both models using BMOA algorithm and the range considered for each parameter is presented in Table 6.

RMSE calculated using the BMOA algorithm is higher than the one calculated with SDA in the case of the SDM, but it is lower than the five parameters method. PSDA for DDM has better results than BMOA for RMSE, MSE, MAE, and MBE, as seen in Table 7. The MAPE and MRE present better results for BMOA algorithm in comparison with all the other algorithms.

The extracted parameters are presented in Appendix A. The highest difference is for reverse saturation current. The reverse saturation current obtained using BMOA is more than two times higher than the one obtained with SDA and more than five times in comparison with 5P.

The current–voltage characteristics measured and calculated are presented in Figure 3a for the SDM model and Figure 3b for the DDM model. The errors calculated for the calculated current are shown in Figure 3c for both models. By analyzing the two current–voltage characteristics for the SDM models, a slight mismatch can be observed around the maximum power, which does not exist for DDM. The SDM presents lower values of the errors around the short circuit current and open circuit voltage than DDM, but the behavior changes drastically around the maximum power.

### 3.3. RTC Photovoltaic Cell

The dataset for the RTC photovoltaic cell [33] is the most commonly used one from the specialized literature to verify the algorithm’s performance for extracting the photovoltaic cell parameters. This cell is old, the shape is circular, with a 57 mm diameter. The current–voltage characteristic was measured at 33 °C under 1000 W/m^2^ illumination. The (I, V) pairs, the calculated current using SMD and DDM models, and the errors are presented in Table 8.

The extracted parameters of the RTC silicon photovoltaic cell using BMOA algorithm are shown for both models in Table 9. Moreover, the considered range for each parameter is presented.

The BMOA algorithm delivers better results for RMSE than PS, ICA, MPCOA, SA, EEGWA, GA, and 5P in the case of the SDM model and PS, ICA, SA, and EEGWA in the case of DDM model. The HSDA and PSDA have better results. The values for all statistical tests are presented in Table 10. The MAPE obtain using BMOA is better for all algorithms except for HSDA, FPA, and CS.

The extracted parameters are presented in Appendix A. There are high differences between the parameters, less for the photogenerated current. These differences can appear due to the algorithms which stop when they find a local minimum.

The current–voltage characteristics measured and calculated are presented in Figure 4a for the SDM model and Figure 4b for the DDM model. The errors calculated for calculated current is presented in Figure 4c for both models. By analyzing the errors, it can be observed that the SDM presents lower values of the errors for two parts of the current–voltage characteristic around the maximum power and open circuit voltage than DDM, but the behavior changes around the short circuit current where SDM has higher errors than DDM.

### 3.4. PWP201 Photovoltaic Panel

The dataset for the PWP201 photovoltaic panel [33] and the calculated current using SMD and DDM models, and the errors are presented in Table 11. It consists of 36 polycrystalline silicon photovoltaic cells connected in series [13]. The current–voltage characteristic was measured at 45 °C under 1000 W/m^2^ illumination.

The extracted parameters of the PWP201 photovoltaic panel using BMOA algorithm are shown for both models in Table 12. Moreover, the considered range for each parameter is presented.

The BMOA algorithm delivers better results for RMSE than PS, FPA, MPCOA, SA, EEGWA, and 5P in the case of the SDM model; PS, CPSO, TVACPSO, and SA in the case of DDM model. The HSDA provides better results. For SDM model, the RMSE is almost equal for GPSO and WDOWOAPSO to the one provided by BMOA. The values for all statistical tests are presented in Table 13. The MAPE obtained using BMOA is better for all algorithms except for HSDA, FPA, and CS.

The extracted parameters are presented in Appendix A. Some differences are observed for the shunt resistance. The results for parameters obtained with EEGWO algorithm are worse in the SDM model. Both ideality factors obtained with BMOA are lower than the ones obtained with HSDA.

The current–voltage characteristics measured and calculated are presented in Figure 5a for the SDM model and Figure 5b for the DDM model. The errors calculated for the calculated current are presented in Figure 5c for both models. By analyzing the errors, it can be observed that there is an alternation between the errors for the two models. SDM presents a lower variation in the errors, between −4 × 10^−03^ and 5 × 10^−03^, than DDM, for which the variation is −8 × 10^−03^ and 6 × 10^−03^. This is reflected in the RMSE, which is higher for the DDM model than SDM model.

### 3.5. Sharp ND-R250A5 Photovoltaic Panel

The dataset for the Sharp ND-R250A5 photovoltaic panel [23] and the calculated current using SMD and DDM models and the errors are presented in Table 14. It consists of 60 polycrystalline silicon photovoltaic cells connected in series [23]. The current–voltage characteristic was measured at 59 °C under 1040 W/m^2^ illumination.

The extracted parameters of the Sharp ND-R250A5 photovoltaic panel using the BMOA algorithm are shown for both models in Table 15. Moreover, the considered range for each parameter is presented.

The BMOA algorithm has lower results for RMSE than HSDA and GCPSO in the case of the SDM and DDM model, and better than 5P in the case of SDM model. The values for all statistical tests are presented in Table 16. The MAPE obtained using BMOA is better for all algorithms.

The extracted parameters are presented in Appendix A. Some differences can be observed for the ideality factor of diode for both models SDM and DDM.

The current–voltage characteristics, measured and calculated, are presented in Figure 6a for the SDM model and Figure 6b for the DDM model. The current voltage calculated has higher values than the one measured for the first half of the curve for SDM model, and the behavior is inverse in the case of the DDM model. The errors obtained for the calculated current are presented in Figure 6c for both models. By analyzing the errors, it can be observed that there is an alternation between the errors for the two models.

### 3.6. Kyocera KC200GT Photovoltaic Panel

The dataset for the Kyocera KC200GT photovoltaic panel [34] and the calculated current using SMD and DDM models and the errors are presented in Table 17. It consists of 54 multicrystalline silicon photovoltaic cells connected in series [13]. The current–voltage characteristic was measured at 25 °C under 1000 W/m^2^ illumination.

The extracted parameters of the Kyocera KC200GT photovoltaic panel using the BMOA algorithm are shown for both models in Table 18. The range considered for each parameter is also presented.

The statistical test shows very good results for BMOA algorithm in comparison with ICA and WDO for both models: SDM and DDM. The HSDA provides better results. The values for all statistical tests are presented in Table 19.

The extracted parameters are presented in Appendix A. There are high differences between the parameters, less for the photogenerated current. These differences can appear due to the algorithms which stop when they find a local minimum.

The current–voltage characteristics measured and calculated are presented in Figure 7a for the SDM model, and Figure 7b for the DDM model. The errors calculated for the calculated current are presented in Figure 7c for both models. There are some mismatches between the current–voltage characteristics measured and calculated for both models. These mismatches lead to the worst results for the statistical tests. For example, the RMSE has the highest values for both models in comparison with the ones for the other five photovoltaic cells and panels. By analyzing the errors, it can be observed that there is an alternation between the errors for the two models for the short circuit current part and maximum power part. For the part of the open circuit voltage, the errors are almost equal, with high variation from −0.39 to 0.566.

### 3.7. Computing Time, Number of Epochs, Population, and Number of Iterations

Two computer configurations were used to extract the parameters of the photovoltaic cells and panels. The first is an HP laptop with AMD Ryzen 5 3550H processor, 8 GB RAM, 2100 Mhz, and 4 GB NVIDIA GeForce GTX 1650, and the second is an Asus Computer with following features: CPU: Intel i9, 10 Core (s), 3.6 GHz 20 MB; GPU: NVIDIA GeForce RTX 3080Ti 12 GB; RAM: 32 GB. The computing time for both configurations and models for each photovoltaic cell and panel is presented in Table 20 for the case with 1000 epochs and 250 populations. The computing time decreases around 4–5 times when the better configuration is used.

The SDA, PSDA, and HSDA algorithms provided some of the best results to extract the parameters of the photovoltaic cells and panels. The time for the SDA and HSDA is almost 5 min, even though the number of iterations is low, 4, using a computer with an 8-thread I7 processor at 1.9 GHz. The time decreases more than six times if parallelization is used, PSDA, the time to end becomes 46 s [12]. In the industrial process, a compromise between the computing time and the accuracy of extracting the parameters must be made for the analysis of the photovoltaic cells [2]. The computing time for the BMOA algorithm decreases more than nine times in comparison with PSDA and more than 30 times against SDA or HSDA.

BMOA algorithm used to extract the parameters of the photovoltaic cells and panels uses 1000 iterations, as seen in Table 1 and Table 21. Most algorithms use more iterations to find the global optimum for the current–voltage characteristic, varying from 100 for the ICA algorithm to 250,000 for the MPCOA algorithm, as seen in Table 21.

The influence of the number of epochs and the population is studied for each photovoltaic cell and panel. There are four cases considered: 1000 epochs with 250, 30 and 15 populations and 500 epochs with 250 populations for each photovoltaic device, and for SDM and DDM models, as seen in Table 22, Table 23, Table 24, Table 25, Table 26 and Table 27.

Comparing the results of the five statistical tests for the two cases, 1000 epochs with 250 populations and 500 epochs with 250 populations, a worsening can be observed if the number of epochs decreases from 1000 to 500, as well as a reduction in the extracting time of the parameters from 1.5 to 2 times.

The variations in the epochs or the population provide weaker results in comparison with the initial case, which considers 1000 epochs and 250 populations for the monocrystalline silicon photovoltaic cell, as seen in Table 22, for both SDM and DDM models.

The decreasing population number leads to improving the RMSE for the SDM model in the case of the amorphous photovoltaic cell. The case 1000 epochs and 15 populations provides the best results for both models, as seen in Table 23.

The case 1000 epochs and 30 populations provides the best results for both models of the RTC photovoltaic cell, as seen in Table 24.

For the SDM model of the PWP201 photovoltaic panel the best results for RMSE are provided by 1000 epochs and 250 populations. For the DDM model, the best results for RMSE are obtained if it uses 1000 epochs and 30 populations, as seen in Table 25.

The same behavior is observed when the population number decreases for the Sharp ND-R250A5 photovoltaic panel as for the amorphous photovoltaic cell, see Table 26.

For Kyocera KC200GT photovoltaic panel an improvement in RMSE is observed only in the case 1000 epochs and 15 populations, for the DDM model. The initial case, 1000 epochs and 250 populations, has the best results, as seen in Table 27.

The calculated values for the current voltage pairs and for the parameters of each photovoltaic cell or panels in the four cases considered are provided in supplementary files, as seen in Appendix A.

## 4. Conclusions

Barnacles mating optimizer algorithm (BMOA) is applied for the first time to extract the intrinsic parameters of three photovoltaic cells: monocrystalline silicon commercial, amorphous silicon, RTC France, PWP201, Sharp ND-R250A5, and Kyocera KC200GT photovoltaic panels, five or seven, in the function of the mathematical model used.

The accuracy provides the performance of the barnacles mating optimizer algorithm in the extraction of the photovoltaic cells and panels parameters. The accuracy of the BMOA algorithm is better than GA and 5P in the case of the monocrystalline silicon photovoltaic cell; 5P in the case of the amorphous silicon photovoltaic cell; PS, ICA, MPCOA, SA, EEGWA, GA, and 5P in the case of the RTC silicon photovoltaic cell; PS, FPA, MPCOA, SA, EEGWA, CPSO, TVACPSO, and 5P, and almost equal for GPSO and WDOWOAPSO in the case of PWP201 photovoltaic panel; 5P in the case of the Sharp ND-R250A5 photovoltaic panel; and ICA and WDO in the case of the Kyocera KC200GT photovoltaic panel.

The barnacles mating optimizer algorithm uses a low number of iterations. This leads to a very short computation time. So, in comparison with the hybrid successive discretization algorithm, the computational time for the barnacles mating optimizer algorithm decreases more than 30 times and 9 times in comparison with parallelized successive discretization algorithm. The reduction in the number of epochs decreases the computational time, but RMSE is worse. Considering the improvement in computational time and worsening in accuracy, this can be a solution for the industrial process. The computational time can be reduced more using the right number for the population with improvements in the results of the statistical tests, especially for RMSE.

All of these make the barnacles mating optimizer algorithm an excellent candidate that can be successfully used to extract the photovoltaic cells and panel parameters.

Although the barnacles mating optimizer algorithm offers very good accuracy for extraction of the photovoltaic cells and panels parameters using the current–voltage characteristic, developing a hybrid algorithm based on BMOA will be the goal for future works. This will lead to an increase in accuracy so that the new hybrid algorithm is among the best. Additionally, these two algorithms will be applied to extract the parameters for the multijunction photovoltaic cells under natural sunlight and in concentrated light.

## Figures and Tables

**Figure 1 sensors-22-06989-f001:**
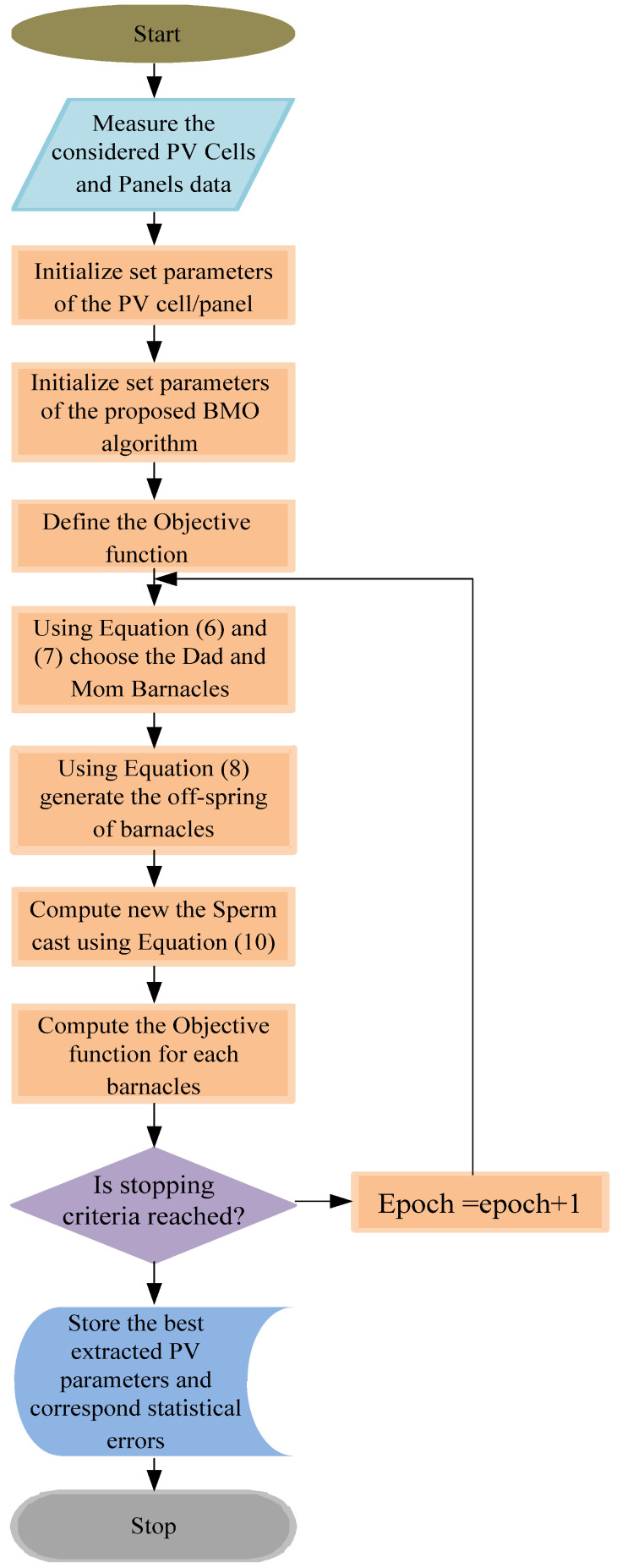
Proposed BMOA-based PV cell and panel parameters extraction pictorial overview.

**Figure 2 sensors-22-06989-f002:**
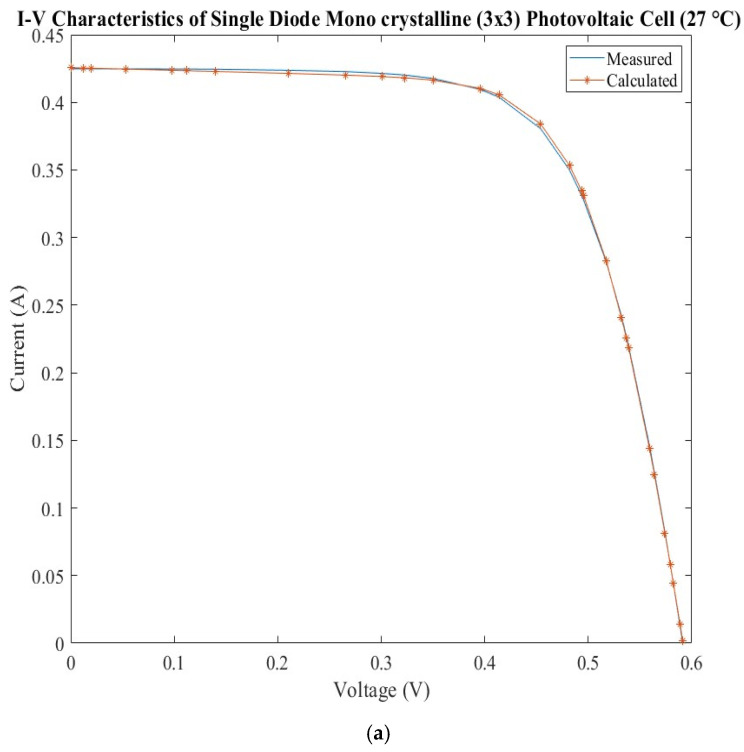
Monocrystalline silicon photovoltaic cell: (**a**) measured and calculated I-V characteristic for SDM; (**b**) measured and calculated I-V characteristic for DDM; (**c**) SDM and DDM errors comparison.

**Figure 3 sensors-22-06989-f003:**
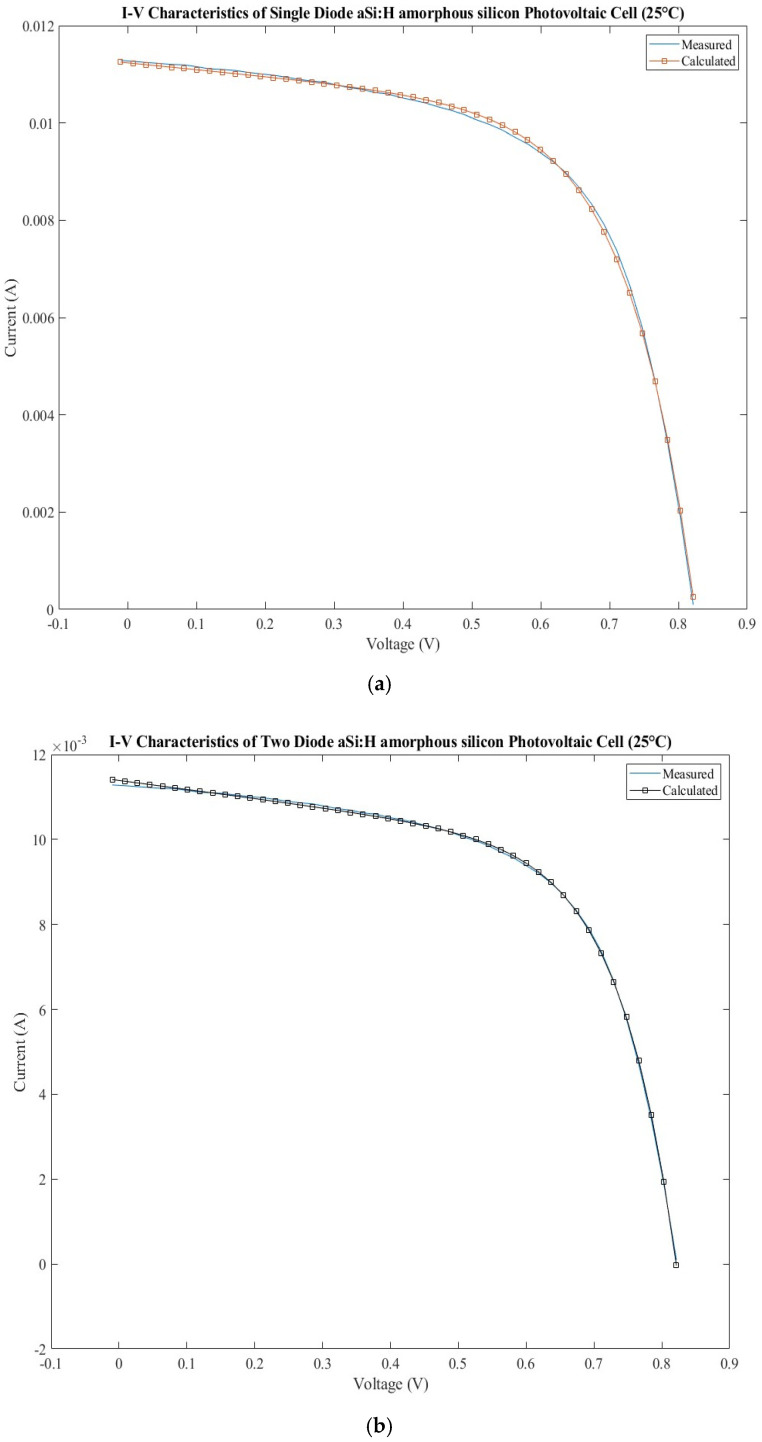
Amorphous silicon photovoltaic cell: (**a**) measured and calculated I-V characteristic for SDM; (**b**) measured and calculated I-V characteristic for DDM; (**c**) SDM and DDM errors comparison.

**Figure 4 sensors-22-06989-f004:**
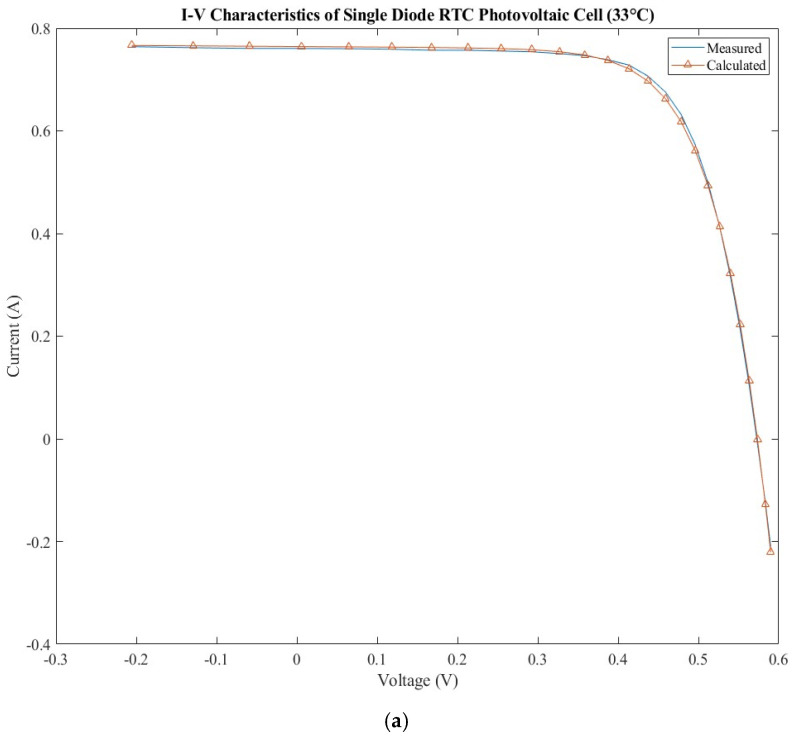
RTC photovoltaic cell: (**a**) measured and calculated I-V characteristic for SDM; (**b**) measured and calculated I-V characteristic for DDM; (**c**) SDM and DDM errors comparison.

**Figure 5 sensors-22-06989-f005:**
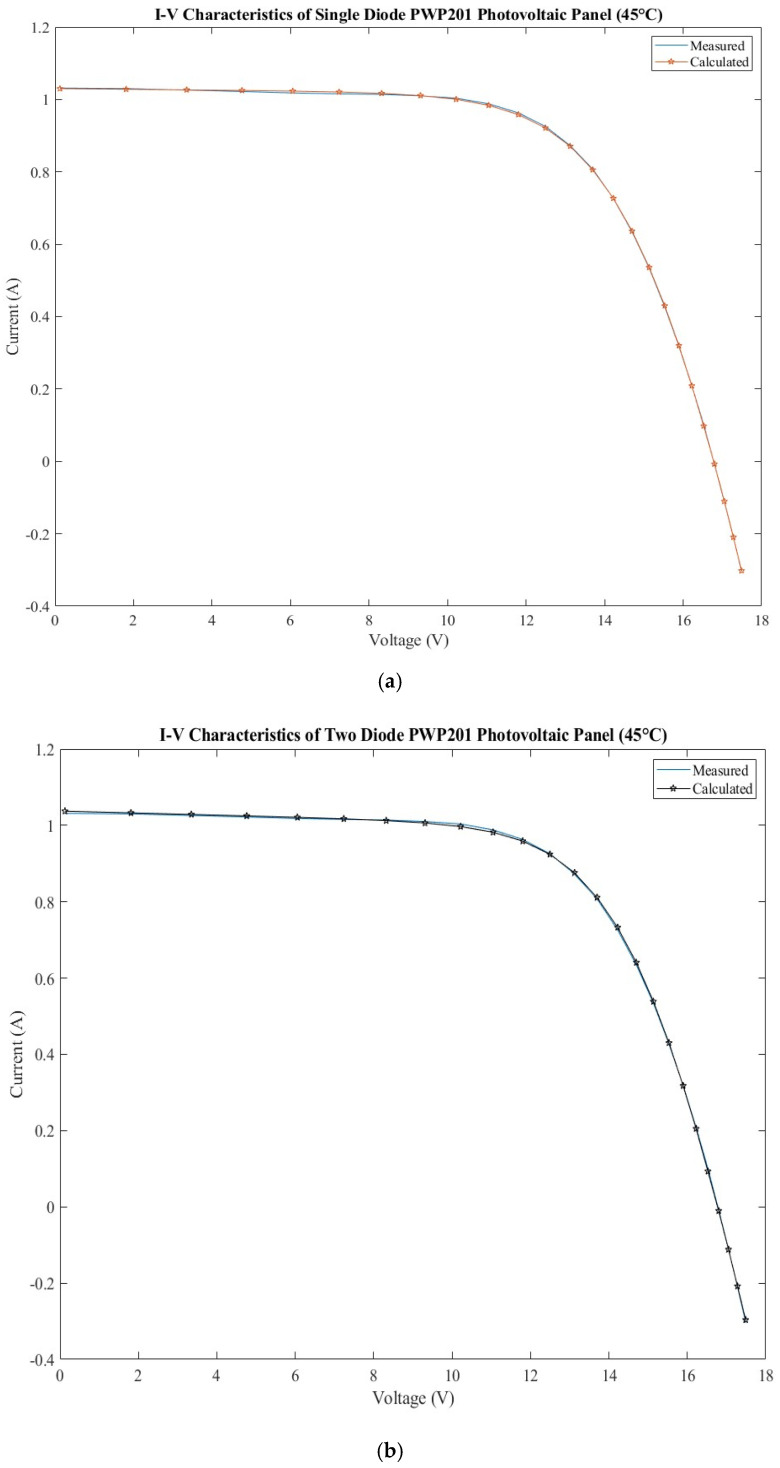
PWP201 photovoltaic panel: (**a**) measured and calculated I-V characteristic for SDM; (**b**) measured and calculated I-V characteristic for DDM; (**c**) SDM and DDM errors comparison.

**Figure 6 sensors-22-06989-f006:**
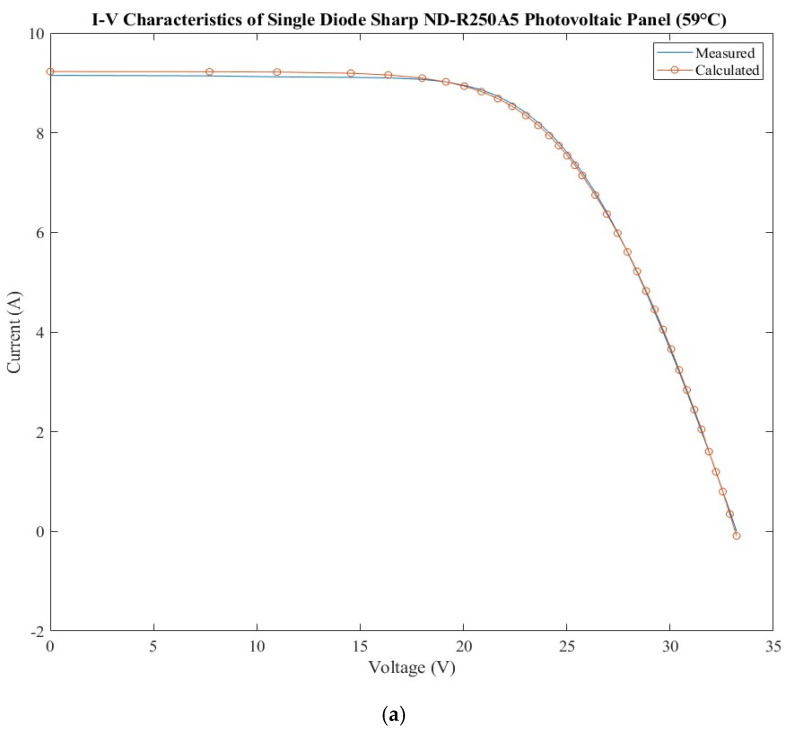
Sharp ND-R250A5 photovoltaic panel: (**a**) measured and calculated I-V characteristic for SDM; (**b**) measured and calculated I-V characteristic for DDM; (**c**) SDM and DDM errors comparison.

**Figure 7 sensors-22-06989-f007:**
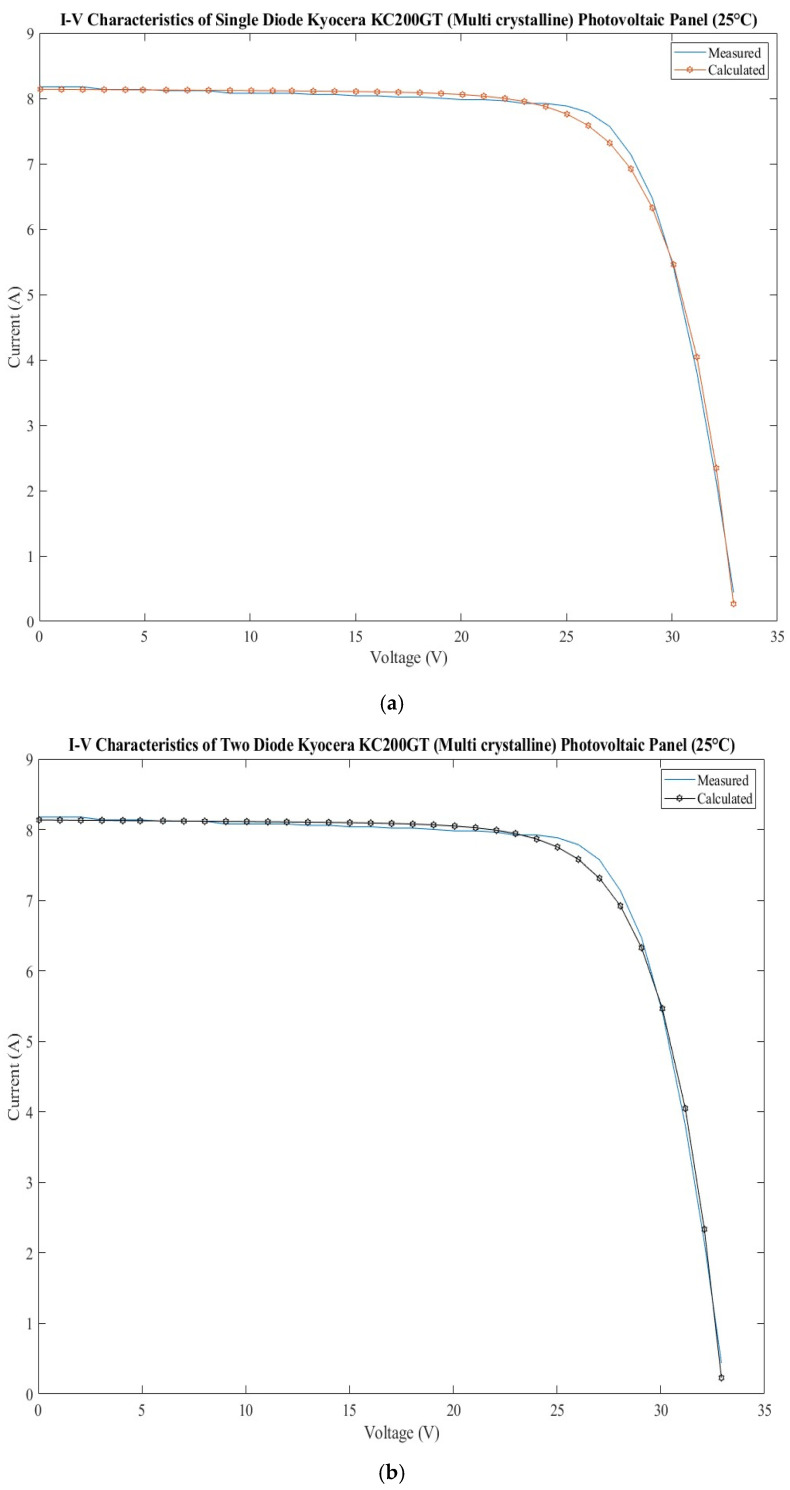
Kyocera KC200GT photovoltaic panel: (**a**) measured and calculated I-V characteristic for SDM; (**b**) measured and calculated I-V characteristic for DDM; (**c**) SDM and DDM errors comparison.

**Table 1 sensors-22-06989-t001:** Proposed BMO algorithm parameters values.

Parameters	Values
Dimension	5 (SDM) and 7 (DDM)
Population	250
length of penis	7
Maximum Epochs	1000

**Table 2 sensors-22-06989-t002:** (I, V) points of monocrystalline silicon photovoltaic cell.

Measured Data		SDM-BMOA	DDM-BMOA	Error Values	
V [V]	I [A]	I_c_ [A]	I_c_ [A]	SDM	DDM
0.0000 × 10^+00^	4.2481 × 10^−01^	4.2565 × 10^−01^	4.2644 × 10^−01^	8.4192 × 10^−04^	1.6346 × 10^−03^
1.1822 × 10^−02^	4.2480 × 10^−01^	4.2542 × 10^−01^	4.2622 × 10^−01^	6.1987 × 10^−04^	1.4246 × 10^−03^
2.0097 × 10^−02^	4.2479 × 10^−01^	4.2525 × 10^−01^	4.2606 × 10^−01^	4.6633 × 10^−04^	1.2795 × 10^−03^
5.3199 × 10^−02^	4.2475 × 10^−01^	4.2459 × 10^−01^	4.2544 × 10^−01^	−1.5208 × 10^−04^	6.9419 × 10^−04^
9.8122 × 10^−02^	4.2465 × 10^−01^	4.2370 × 10^−01^	4.2459 × 10^−01^	−9.5077 × 10^−04^	−6.1840 × 10^−05^
1.1172 × 10^−01^	4.2464 × 10^−01^	4.2343 × 10^−01^	4.2433 × 10^−01^	−1.2162 × 10^−03^	−3.1527 × 10^−04^
1.4009 × 10^−01^	4.2448 × 10^−01^	4.2286 × 10^−01^	4.2378 × 10^−01^	−1.6213 × 10^−03^	−6.9753 × 10^−04^
2.0984 × 10^−01^	4.2372 × 10^−01^	4.2143 × 10^−01^	4.2238 × 10^−01^	−2.2875 × 10^−03^	−1.3345 × 10^−03^
2.6599 × 10^−01^	4.2275 × 10^−01^	4.2013 × 10^−01^	4.2103 × 10^−01^	−2.6248 × 10^−03^	−1.7180 × 10^−03^
3.0087 × 10^−01^	4.2140 × 10^−01^	4.1907 × 10^−01^	4.1987 × 10^−01^	−2.3300 × 10^−03^	−1.5243 × 10^−03^
3.2274 × 10^−01^	4.2019 × 10^−01^	4.1815 × 10^−01^	4.1885 × 10^−01^	−2.0398 × 10^−03^	−1.3461 × 10^−03^
3.5052 × 10^−01^	4.1759 × 10^−01^	4.1642 × 10^−01^	4.1689 × 10^−01^	−1.1729 × 10^−03^	−6.9870 × 10^−04^
3.9604 × 10^−01^	4.0947 × 10^−01^	4.1026 × 10^−01^	4.1013 × 10^−01^	7.9042 × 10^−04^	6.5952 × 10^−04^
4.1377 × 10^−01^	4.03773× 10^−01^	4.0554 × 10^−01^	4.0509 × 10^−01^	1.7659 × 10^−03^	1.3160 × 10^−03^
4.5396 × 10^−01^	3.8056 × 10^−01^	3.8404 × 10^−01^	3.8285 × 10^−01^	3.4856 × 10^−03^	2.2960 × 10^−03^
4.8175 × 10^−01^	3.5006 × 10^−01^	3.5345 × 10^−01^	3.5203 × 10^−01^	3.3869 × 10^−03^	1.9718 × 10^−03^
4.9357 × 10^−01^	3.3110 × 10^−01^	3.3461 × 10^−01^	3.3326 × 10^−01^	3.5158 × 10^−03^	2.1644 × 10^−03^
4.9534 × 10^−01^	3.28027× 10^−01^	3.3141 × 10^−01^	3.3008 × 10^−01^	3.3799 × 10^−03^	2.0497 × 10^−03^
5.1721 × 10^−01^	2.8237 × 10^−01^	2.8310 × 10^−01^	2.8227 × 10^−01^	7.2819 × 10^−04^	−9.3791 × 10^−05^
0.531988	2.4221 × 10^−01^	2.4082 × 10^−01^	2.4055 × 10^−01^	−1.3918 × 10^−03^	−1.6566 × 10^−03^
5.3672 × 10^−01^	2.2709 × 10^−01^	2.2595 × 10^−01^	2.2586 × 10^−01^	−1.1374 × 10^−03^	−1.2254 × 10^−03^
5.3908 × 10^−01^	2.1931 × 10^−01^	2.1818 × 10^−01^	2.1819 × 10^−01^	−1.1230 × 10^−03^	−1.1200 × 10^−03^
5.5918 × 10^−01^	1.4596 × 10^−01^	1.4393 × 10^−01^	1.4464 × 10^−01^	−2.0329 × 10^−03^	−1.3230 × 10^−03^
5.6391 × 10^−01^	1.2639 × 10^−01^	1.2478 × 10^−01^	1.2558 × 10^−01^	−1.6091 × 10^−03^	−8.0630 × 10^−04^
5.7396 × 10^−01^	8.2714 × 10^−02^	8.1602 × 10^−02^	8.2486 × 10^−02^	−1.1123 × 10^−03^	−2.2824 × 10^−04^
5.7928 × 10^−01^	5.7940 × 10^−02^	5.7986 × 10^−02^	5.8795 × 10^−02^	4.6065 × 10^−05^	8.5527 × 10^−04^
5.8223 × 10^−01^	4.4104 × 10^−02^	4.4240 × 10^−02^	4.4994 × 10^−02^	1.3637 × 10^−04^	8.8971 × 10^−04^
5.8873 × 10^−01^	1.2269 × 10^−02^	1.3779 × 10^−02^	1.4279 × 10^−02^	1.5098 × 10^−03^	2.0104 × 10^−03^
5.9121 × 10^−01^	0.0000 × 10^+00^	1.7083 × 10^−03^	2.0874 × 10^−03^	1.7083 × 10^−03^	2.0874 × 10^−03^

**Table 3 sensors-22-06989-t003:** Monocrystalline silicon photovoltaic cell parameters and range.

Algorithm	I_ph_ [A]	I_o1_ [A]	n_1_	R_s_ [Ω]	R_sh_ [Ω]	I_o2_ [A]	n_2_
Range Set SMD	0–1	1 × 10^−12^ – 1 × 10^−05^	1–2	0–1	0–200		
BMOA SMD	4.2660 × ^10−01^	5.4453 × 10^−08^	1.4429 × 10^+00^	1.1186 × 10^−01^	5.0305 × 10^+01^		
Range Set DDM	0–1	1 × 10^−12^– 1× 10^−05^	1–2	0–1	0–200	1 × 10^−12^– 1 × 10^−05^	1–3
BMOA DDM	4.2731 × 10^−01^	5.6095 × 10^−08^	1.4532 × 10^+00^	1.0822 × 10^−01^	5.3106 × 10^+01^	5.3674 × 10^−07^	2.0685 × 10^+00^

**Table 4 sensors-22-06989-t004:** Monocrystalline silicon photovoltaic cell (SDM and DDM) statistical tests.

Algorithms	RMSE	MAPE	MSE	MAE	MBE	MRE	Model
**BMOA**	**1.8456 × 10^−03^**	**5.1804 × 10^−01^**	**3.4062 × 10^−06^**	**1.5580 × 10^−03^**	**5.1804 × 10^−01^**	**5.1804 × 10^−03^**	SDM
SDA	5.6309 × 10^−04^	3.90148	3.94877 × 10^−07^	4.98157 × 10^−04^	1.75553 × 10^−04^	3.90148 × 10^−02^	
GA	2.1244 × 10^−03^	7.37113	5.40299 × 10^−06^	1.60728 × 10^−03^	1.38046 × 10^−03^	7.37113 × 10^−02^	
5P	2.2469 × 10^−03^	4.97806	5.04863 × 10^−06^	1.63766 × 10^−03^	2.13079 × 10^−04^	4.97806 × 10^−02^	
**BMOA**	**1.3754 × 10^−03^**	**4.0682 × 10^−01^**	**1.8918 × 10^−06^**	**1.2235 × 10^−03^**	**2.4771 × 10^−04^**	**4.0682 × 10^−03^**	DDM
PSDA	5.4057 × 10^−04^	3.76948	2.9221 × 10^−07^	4.6859 × 10^−04^	−5.7078 × 10^−07^	3.74948 × 10^−02^	

**Table 5 sensors-22-06989-t005:** (I, V) points of aSi photovoltaic cell.

Measured Data		SDM-BMOA	DDM-BMOA	Error Values	
V [V]	I [A]	I_c_ [A]	I_c_ [A]	SDM	DDM
−1.0000 × 10^−02^	1.1283 × 10^−02^	1.1249 × 10^−02^	1.1410 × 10^−02^	−3.4491 × 10^−05^	1.2723 × 10^−04^
8.4800 × 10^−03^	1.1266 × 10^−02^	1.1223 × 10^−02^	1.1371 × 10^−02^	−4.2674 × 10^−05^	1.0548 × 10^−04^
2.6955 × 10^−02^	1.1243 × 10^−02^	1.1197 × 10^−02^	1.1332 × 10^−02^	−4.5323 × 10^−05^	8.9324 × 10^−05^
4.5435 × 10^−02^	1.1224 × 10^−02^	1.1171 × 10^−02^	1.1293 × 10^−02^	−5.2267 × 10^−05^	6.8920 × 10^−05^
6.3915 × 10^−02^	1.1202 × 10^−02^	1.1145 × 10^−02^	1.1253 × 10^−02^	−5.7019 × 10^−05^	5.0773 × 10^−05^
8.2390 × 10^−02^	1.1193 × 10^−02^	1.1119 × 10^−02^	1.1214 × 10^−02^	−7.3795 × 10^−05^	2.0682 × 10^−05^
1.0087 × 10^−01^	1.1155 × 10^−02^	1.1093 × 10^−02^	1.1174 × 10^−02^	−6.1637 × 10^−05^	1.9614 × 10^−05^
1.1935 × 10^−01^	1.1110 × 10^−02^	1.1067 × 10^−02^	1.1135 × 10^−02^	−4.3771 × 10^−05^	2.4362 × 10^−05^
1.3783 × 10^−01^	1.1099 × 10^−02^	1.1040 × 10^−02^	1.1095 × 10^−02^	−5.9032 × 10^−05^	−3.8842 × 10^−06^
1.5631 × 10^−01^	1.1078 × 10^−02^	1.1013 × 10^−02^	1.1055 × 10^−02^	−6.5183 × 10^−05^	−2.2872 × 10^−05^
1.7479 × 10^−01^	1.1040 × 10^−02^	1.0986 × 10^−02^	1.1015 × 10^−02^	−5.3980 × 10^−05^	−2.4322 × 10^−05^
1.9326 × 10^−01^	1.1007 × 10^−02^	1.0958 × 10^−02^	1.0975 × 10^−02^	−4.8988 × 10^−05^	−3.1766 × 10^−05^
2.1175 × 10^−01^	1.0982 × 10^−02^	1.0930 × 10^−02^	1.0935 × 10^−02^	−5.1826 × 10^−05^	−4.6797 × 10^−05^
2.3020 × 10^−01^	1.0941 × 10^−02^	1.0901 × 10^−02^	1.0894 × 10^−02^	−4.0319 × 10^−05^	−4.7156 × 10^−05^
2.4870 × 10^−01^	1.0896 × 10^−02^	1.0871 × 10^−02^	1.0853 × 10^−02^	−2.4839 × 10^−05^	−4.3223 × 10^−05^
2.6715 × 10^−01^	1.0861 × 10^−02^	1.0841 × 10^−02^	1.0811 × 10^−02^	−2.0087 × 10^−05^	−4.9571 × 10^−05^
2.8565 × 10^−01^	1.0838 × 10^−02^	1.0809 × 10^−02^	1.0769 × 10^−02^	−2.8222 × 10^−05^	−6.8351 × 10^−05^
3.0415 × 10^−01^	1.0777 × 10^−02^	1.0776 × 10^−02^	1.0726 × 10^−02^	−1.0002 × 10^−06^	−5.1202 × 10^−05^
3.2260 × 10^−01^	1.0724 × 10^−02^	1.0742 × 10^−02^	1.0682 × 10^−02^	1.8287 × 10^−05^	−4.1297 × 10^−05^
3.4110 × 10^−01^	1.0690 × 10^−02^	1.0705 × 10^−02^	1.0637 × 10^−02^	1.5603 × 10^−05^	−5.2613 × 10^−05^
3.5955 × 10^−01^	1.0624 × 10^−02^	1.0666 × 10^−02^	1.0590 × 10^−02^	4.2625 × 10^−05^	−3.3302 × 10^−05^
3.7805 × 10^−01^	1.0596 × 10^−02^	1.0624 × 10^−02^	1.0542 × 10^−02^	2.8527 × 10^−05^	−5.4090 × 10^−05^
3.9650 × 10^−01^	1.0523 × 10^−02^	1.0579 × 10^−02^	1.0491 × 10^−02^	5.6515 × 10^−05^	−3.1578 × 10^−05^
4.1500 × 10^−01^	1.0467 × 10^−02^	1.0529 × 10^−02^	1.0437 × 10^−02^	6.2285 × 10^−05^	−2.9921 × 10^−05^
4.3350 × 10^−01^	1.0407 × 10^−02^	1.0474 × 10^−02^	1.0380 × 10^−02^	6.7134 × 10^−05^	−2.7611 × 10^−05^
4.5195 × 10^−01^	1.0327 × 10^−02^	1.0413 × 10^−02^	1.0318 × 10^−02^	8.6117 × 10^−05^	−9.3793 × 10^−06^
4.7045 × 10^−01^	1.0258 × 10^−02^	1.0344 × 10^−02^	1.0250 × 10^−02^	8.6847 × 10^−05^	−7.3914 × 10^−06^
4.8890 × 10^−01^	1.0173 × 10^−02^	1.0266 × 10^−02^	1.0176 × 10^−02^	9.3758 × 10^−05^	3.0147 × 10^−06^
5.0740 × 10^−01^	1.0060 × 10^−02^	1.0177 × 10^−02^	1.0092 × 10^−02^	1.1683 × 10^−04^	3.2075 × 10^−05^
5.2585 × 10^−01^	9.9670 × 10^−03^	1.0074 × 10^−02^	9.9975 × 10^−03^	1.0653 × 10^−04^	3.0455 × 10^−05^
5.4435 × 10^−01^	9.8534 × 10^−03^	9.9534 × 10^−03^	9.8890 × 10^−03^	1.0004 × 10^−04^	3.5600 × 10^−05^
5.6285 × 10^−01^	9.7021 × 10^−03^	9.8131 × 10^−03^	9.7633 × 10^−03^	1.1096 × 10^−04^	6.1221 × 10^−05^
5.8130 × 10^−01^	9.5669 × 10^−03^	9.6485 × 10^−03^	9.6166 × 10^−03^	8.1613 × 10^−05^	4.9667 × 10^−05^
5.9980 × 10^−01^	9.3838 × 10^−03^	9.4536 × 10^−03^	9.4426 × 10^−03^	6.9832 × 10^−05^	5.8811 × 10^−05^
6.1825 × 10^−01^	9.2011 × 10^−03^	9.2231 × 10^−03^	9.2358 × 10^−03^	2.1997 × 10^−05^	3.4660 × 10^−05^
6.3675 × 10^−01^	8.9748 × 10^−03^	8.9479 × 10^−03^	8.9866 × 10^−03^	−2.6909 × 10^−05^	1.1784 × 10^−05^
6.5520 × 10^−01^	8.6853 × 10^−03^	8.6201 × 10^−03^	8.6861 × 10^−03^	−6.5196 × 10^−05^	7.9574 × 10^−07^
6.7370 × 10^−01^	8.3376 × 10^−03^	8.2265 × 10^−03^	8.3197 × 10^−03^	−1.1109 × 10^−04^	−1.7878 × 10^−05^
6.9215 × 10^−01^	7.9110 × 10^−03^	7.7554 × 10^−03^	7.8733 × 10^−03^	−1.5561 × 10^−04^	−3.7660 × 10^−05^
7.1065 × 10^−01^	7.3725 × 10^−03^	7.1873 × 10^−03^	7.3244 × 10^−03^	−1.8519 × 10^−04^	−4.8138 × 10^−05^
7.2915 × 10^−01^	6.6650 × 10^−03^	6.5029 × 10^−03^	6.6486 × 10^−03^	−1.6210 × 10^−04^	−1.6356 × 10^−05^
7.4760 × 10^−01^	5.7870 × 10^−03^	5.6797 × 10^−03^	5.8172 × 10^−03^	−1.0733 × 10^−04^	3.0178 × 10^−05^
7.6610 × 10^−01^	4.7014 × 10^−03^	4.6830 × 10^−03^	4.7862 × 10^−03^	−1.8428 × 10^−05^	8.4818 × 10^−05^
7.8455 × 10^−01^	3.4034 × 10^−03^	3.4816 × 10^−03^	3.5125 × 10^−03^	7.8245 × 10^−05^	1.0913 × 10^−04^
8.0305 × 10^−01^	1.8769 × 10^−03^	2.0247 × 10^−03^	1.9281 × 10^−03^	1.4777 × 10^−04^	5.1176 × 10^−05^
8.2150 × 10^−01^	9.5797 × 10^−05^	2.6607 × 10^−04^	−3.4625 × 10^−05^	1.7027 × 10^−04^	−1.3042 × 10^−04^

**Table 6 sensors-22-06989-t006:** Amorphous silicon photovoltaic cell parameters and range.

Algorithm	I_ph_ [A]	I_o1_ [A]	n_1_	R_s_ [Ω]	R_sh_ [Ω]	I_o2_ [A]	n_2_
Range Set SDM	0–0.1	1 × 10^−12^– 1 × 10^−05^	1–5	0–0.5	0–1000		
BMOA SDM	1.1235 × 10^−02^	1.7698 × 10^−06^	3.7077 × 10^+00^	0.0000 × 10^+00^	7.2923 × 10^+02^		
Range Set DDM	0–0.1	1 × 10^−12^ – 1 × 10^−05^	1–5	0–0.5	0–1000	1 × 10^−12^– 1 × 10^−05^	1–5
BMOA DDM	1.1389 × 10^−02^	5.0617 × 10^−07^	3.2431 × 10^+00^	0.0000 × 10^+00^	4.7227 × 10^+02^	1.0000 × 10^−12^	3.8000 × 10^+00^

**Table 7 sensors-22-06989-t007:** Amorphous silicon photovoltaic cell (SDM and DDM) statistical tests.

Algorithms	RMSE	MAPE	MSE	MAE	MBE	MRE	Model
**BMOA**	**8.2115 × 10^−05^**	**7.3431 × 10^−01^**	**6.7430 × 10^−09^**	**6.9524 × 10^−05^**	**−1.6201 × 10^−06^**	**7.3431 × 10^−03^**	SDM
SDA	4.6330 × 10^−05^	4.3431	2.1464 × 10^−09^	3.9830 × 10−05	−4.3943 × 10^−07^	9.8567 × 10^−03^	
5P	2.0863 × 10^−04^	5.453	4.3526 × 10^−08^	1.8995 × 10^−04^	−1.44047 × 10^−04^	1.2441 × 10^−02^	
**BMOA**	**5.3511 × 10^−05^**	**4.6531 × 10^−01^**	**2.8634 × 10^−09^**	**4.4055 × 10^−05^**	**3.7606 × 10^−06^**	**4.6531 × 10^−03^**	DDM
PSDA	4.7973 × 10^−05^	3.24501	2.3014 × 10^−09^	3.9287 × 10^−05^	2.82013 × 10^−08^	3.5645 × 10^−03^	

**Table 8 sensors-22-06989-t008:** (I, V) points of RTC silicon photovoltaic cell.

Measured Data		SDM-BMOA	DDM-BMOA	Error Values	
V [V]	I [A]	I_c_ [A]	I_c_ [A]	SDM	DDM
−0.2057	7.6400 × 10^−01^	7.6632 × 10^−01^	7.6451 × 10^−01^	2.3244 × 10^−03^	5.1434 × 10^−04^
−0.1291	7.6200 × 10^−01^	7.6556 × 10^−01^	7.6375 × 10^−01^	3.5583 × 10^−03^	1.7481 × 10^−03^
−0.0588	7.6050 × 10^−01^	7.6485 × 10^−01^	7.6304 × 10^−01^	4.3536 × 10^−03^	2.5438 × 10^−03^
5.7000 × 10^−03^	7.6050 × 10^−01^	7.6420 × 10^−01^	7.6239 × 10^−01^	3.7015 × 10^−03^	1.8941 × 10^−03^
6.4600 × 10^−02^	7.6000 × 10^−01^	7.6359 × 10^−01^	7.6179 × 10^−01^	3.5901 × 10^−03^	1.7900 × 10^−03^
1.1850 × 10^−01^	7.5900 × 10^−01^	7.6299 × 10^−01^	7.6121 × 10^−01^	3.9870 × 10^−03^	2.2062 × 10^−03^
1.6780 × 10^−01^	7.5700 × 10^−01^	7.6233 × 10^−01^	7.6059 × 10^−01^	5.3273 × 10^−03^	3.5912 × 10^−03^
2.1320 × 10^−01^	7.5700 × 10^−01^	7.6147 × 10^−01^	7.5983 × 10^−01^	4.4712 × 10^−03^	2.8287 × 10^−03^
2.5450 × 10^−01^	7.5550 × 10^−01^	7.6018 × 10^−01^	7.5871 × 10^−01^	4.6790 × 10^−03^	3.2120 × 10^−03^
2.9240 × 10^−01^	7.5400 × 10^−01^	7.5800 × 10^−01^	7.5683 × 10^−01^	3.9993 × 10^−03^	2.8340 × 10^−03^
3.2690 × 10^−01^	7.5050 × 10^−01^	7.5425 × 10^−01^	7.5355 × 10^−01^	3.7461 × 10^−03^	3.0489 × 10^−03^
3.5850 × 10^−01^	7.4650 × 10^−01^	7.4784 × 10^−01^	7.4780 × 10^−01^	1.3351 × 10^−03^	1.3008 × 10^−03^
3.8730 × 10^−01^	7.3850 × 10^−01^	7.3735 × 10^−01^	7.3815 × 10^−01^	−1.1528 × 10^−03^	−3.4621 × 10^−04^
4.1370 × 10^−01^	7.2800 × 10^−01^	7.2080 × 10^−01^	7.2255 × 10^−01^	−7.1983 × 10^−03^	−5.4485 × 10^−03^
4.3730 × 10^−01^	7.0650 × 10^−01^	6.9679 × 10^−01^	6.9943 × 10^−01^	−9.7054 × 10^−03^	−7.0685 × 10^−03^
4.5900 × 10^−01^	6.7550 × 10^−01^	6.6266 × 10^−01^	6.6596 × 10^−01^	−1.2835 × 10^−02^	−9.5383 × 10^−03^
4.7840 × 10^−01^	6.3200 × 10^−01^	6.1789 × 10^−01^	6.2144 × 10^−01^	−1.4108 × 10^−02^	−1.0561 × 10^−02^
4.9600 × 10^−01^	5.7300 × 10^−01^	5.6136 × 10^−01^	5.6469 × 10^−01^	−1.1642 × 10^−02^	−8.3085 × 10^−03^
5.1190 × 10^−01^	4.9900 × 10^−01^	4.9347 × 10^−01^	4.9619 × 10^−01^	−5.5290 × 10^−03^	−2.8057 × 10^−03^
5.2650 × 10^−01^	4.1300 × 10^−01^	4.1345 × 10^−01^	4.1526 × 10^−01^	4.4992 × 10^−04^	2.2593 × 10^−03^
5.3980 × 10^−01^	3.1650 × 10^−01^	3.2306 × 10^−01^	3.2392 × 10^−01^	6.5588 × 10^−03^	7.4193 × 10^−03^
5.5210 × 10^−01^	2.1200 × 10^−01^	2.2195 × 10^−01^	2.2203 × 10^−01^	9.9539 × 10^−03^	1.0032 × 10^−02^
5.6330 × 10^−01^	1.0350 × 10^−01^	1.1319 × 10^−01^	1.1292 × 10^−01^	9.6867 × 10^−03^	9.4162 × 10^−03^
5.7360 × 10^−01^	−0.0100	−1.8939 × 10^−03^	−1.7061 × 10^−03^	8.1061 × 10^−03^	8.2939 × 10^−03^
5.8330 × 10^−01^	−0.1230	−1.2673 × 10^−01^	−1.2551 × 10^−01^	−3.7317 × 10^−03^	−2.5088 × 10^−03^
5.9000 × 10^−01^	−0.2100	−2.2114 × 10^−01^	−2.1823 × 10^−01^	−1.1140 × 10^−02^	−8.2341 × 10^−03^

**Table 9 sensors-22-06989-t009:** RTC silicon photovoltaic cell parameters and range.

Algorithm	I_ph_ [A]	I_o1_ [A]	n_1_	R_s_ [Ω]	R_sh_ [Ω]	I_o2_ [A]	n_2_
Range Set SDM	0–1	1 × 10^−12^– 1 × 10^−05^	1–2	0–0.5	0–100		
BMOA SDM	7.6441 × 10^−01^	6.3129 × 10^−06^	1.8578 × 10 ^+00^	1.9959 × 10^−02^	1.0000 × 10^+02^		
Range Set DDM	0–1	1 × 10^−12^– 1 × 10^−05^	1–2	0–0.5	0–100	1 × 10^−12^– 1 × 10^−05^	1–2.5
BMOA DDM	7.6263 × 10^−01^	1.0000 × 10^−12^	2.0000 × 10^+00^	2.3399 × 10^−02^	9.9937 × 10^+01^	3.5301 × 10^−06^	1.7701 × 10^+00^

**Table 10 sensors-22-06989-t010:** RTC silicon photovoltaic cell (SDM and DDM) statistical tests.

Algorithms	RMSE	MAPE	MSE	MAE	MBE	MRE	Model
**BMOA**	**7.0675 × 10^−03^**	**1.0939 × 10+00**	**4.9949 × 10^−05^**	**6.0335 × 10^−03^**	**1.0712 × 10^−04^**	**1.0939 × 10^−02^**	SDM
HSDA	9.8602 × 10^−04^	7.39228 × 10^−01^	9.7223 × 10^−07^	8.2795 × 10^−03^	1.42971 × 10^−08^	7.3922 × 10^−02^	
PS	1.4936 × 10^−02^	1.42489 × 10^+01^	2.2309 × 10^−04^	9.38345 × 10^−03^	8.44223 × 10^−03^	1.42489 × 10^−01^	
ICA	1.1581 × 10^−01^	30.0354	1.34134 × 10^−02^	6.47661 × 10^−02^	−6.3638 × 10^−02^	3.00354 × 10^−01^	
CPSO	1.3860 × 10^−03^	1.63759 × 10^+00^	1.92121 × 10^−06^	1.08192 × 10^−03^	−4.97143 × 10^−04^	1.63759 × 10^−02^	
MPCOA	1.3249 × 10^−02^	6.31312	1.75534 × 10^−04^	7.76018 × 10^−03^	7.4493 × 10^−03^	6.31312 × 10^−02^	
SA	1.8998 × 10^−02^	1.93252 × 10^+01^	3.61 × 10^−04^	1.13907 × 10^−02^	1.07684 × 10^−02^	1.93252 × 10^−01^	
FPA	1.2188 × 10^−03^	4.41566 × 10^−01^	1.48547 × 10^−06^	8.3904 × 10^−04^	4.01361 × 10^−04^	4.41566 × 10^−03^	
CS	1.1632 × 10^−03^	3.11765 × 10^−01^	1.35314 × 10^−06^	7.99007 × 10^−04^	3.36253 × 10^−04^	3.11765 × 10^−03^	
EEGWO	3.6492 × 10^−02^	2.3744 × 10^+01^	1.41512 × 10^−03^	3.69366 × 10^−02^	3.69366 × 10^−02^	2.37447 × 10^−01^	
GA	1.908 × 10^−02^	7.37113	5.40299 × 10^−06^	1.60728 × 10^−03^	1.38046 × 10^−03^	7.37113 × 10^−02^	
5P	9.134 × 10^−03^	1.97806	4.04863 × 10^−06^	6.13732 × 10^−03^	3.60061 × 10^−03^	1.97806 × 10^−02^	
**BMOA**	**5.6269 × 10^−03^**	**8.3504 × 10^−01^**	**3.1662 × 10^−05^**	**4.6059 × 10^−03^**	**3.8898 × 10^−04^**	**8.3504 × 10^−03^**	DDM
PSDA	9.8247 × 10^−04^	3.76948 × 10^−01^	2.9221 × 10^−07^	4.6859 × 10^−04^	−5.7078 × 10^−07^	3.74948 × 10^−02^	
PS	1.5229 × 10−^02^	6.77472 × 10^+00^	2.3189 × 10^−04^	9.62646 × 10^−03^	9.35489 × 10^−03^	6.77472 × 10^−02^	
ICA	1.0631 × 10^−01^	2.76652 × 10^+01^	1.13012 × 10^−02^	4.55085 × 10^−02^	4.2916 × 10^−02^	2.76652 × 10^−01^	
MPCOA	2.3349 × 10^−03^	1.58837 × 10^+00^	5.45149 × 10^−06^	1.56851 × 10^−03^	1.16253 × 10^−03^	1.58837 × 10^−02^	
SA	1.6644 × 10^−02^	7.30052 × 10^+00^	2.7701 × 10^−04^	1.00216 × 10^−02^	9.24536 × 10^−03^	7.30052 × 10^−02^	
FPA	2.0978 × 10^−03^	1.36911 × 10^+00^	4.4006 × 10^−06^	1.38208 × 10^−03^	1.03111 × 10^−03^	1.36911 × 10^−02^	
EEGWO	3.0563 × 10^−02^	1.51965 × 10^+01^	9.34099 × 10^−04^	2.90363 × 10^−02^	1.52906 × 10^−02^	1.51965 × 10^−01^	

**Table 11 sensors-22-06989-t011:** (I, V) points of the PWP201 photovoltaic panel.

Measured Data		SDM-BMOA	DDM-BMOA	Error Values	
V [V]	I [A]	I_c_ [A]	I_c_ [A]	SDM	DDM
1.2480 × 10^−01^	1.0315 × 10^+00^	1.0298 × 10^+00^	1.0372 × 10^+00^	−1.6515 × 10^−03^	5.6855 × 10^−03^
1.8093 × 10^+00^	1.0300 × 10^+00^	1.0281 × 10^+00^	1.0327 × 10^+00^	−1.8639 × 10^−03^	2.6821 × 10^−03^
3.3511 × 10^+00^	1.0260 × 10^+00^	1.0265 × 10^+00^	1.0285 × 10^+00^	5.1222 × 10^−04^	2.5431 × 10^−03^
4.7622 × 10^+00^	1.0220 × 10^+00^	1.0249 × 10^+00^	1.0247 × 10^+00^	2.8745 × 10^−03^	2.6802 × 10^−03^
6.0538 × 10^+00^	1.0180 × 10^+00^	1.0230 × 10^+00^	1.0209 × 10^+00^	5.0233 × 10^−03^	2.9480 × 10^−03^
7.2364 × 10^+00^	1.0155 × 10^+00^	1.0206 × 10^+00^	1.0171 × 10^+00^	5.0719 × 10^−03^	1.5568 × 10^−03^
8.3189 × 10^+00^	1.0140 × 10^+00^	1.0168 × 10^+00^	1.0125 × 10^+00^	2.8334 × 10^−03^	−1.5326 × 10^−03^
9.3097 × 10^+00^	1.0100 × 10^+00^	1.0107 × 10^+00^	1.0062 × 10^+00^	6.9053 × 10^−04^	−3.7652 × 10^−03^
1.0216 × 10^+01^	1.0035 × 10^+00^	1.0004 × 10^+00^	9.9678 × 10^−01^	−3.0664 × 10^−03^	−6.7241 × 10^−03^
1.1045 × 10^+01^	9.8800 × 10^−01^	9.8387 × 10^−01^	9.8189 × 10^−01^	−4.1262 × 10^−03^	−6.1052 × 10^−03^
1.1802 × 10^+01^	9.6300 × 10^−01^	9.5834 × 10^−01^	9.5869 × 10^−01^	−4.6554 × 10^−03^	−4.3061 × 10^−03^
1.2493 × 10^+01^	9.2550 × 10^−01^	9.2124 × 10^−01^	9.2410 × 10^−01^	−4.2601 × 10^−03^	−1.4018 × 10^−03^
1.3123 × 10^+01^	8.7250 × 10^−01^	8.7077 × 10^−01^	8.7572 × 10^−01^	−1.7319 × 10^−03^	3.2241 × 10^−03^
1.3698 × 10^+01^	8.0750 × 10^−01^	8.0550 × 10^−01^	8.1154 × 10^−01^	−2.0036 × 10^−03^	4.0432 × 10^−03^
1.4222 × 10^+01^	7.2650 × 10^−01^	7.2689 × 10^−01^	7.3283 × 10^−01^	3.8854 × 10^−04^	6.3268 × 10^−03^
1.4700 × 10^+01^	6.3450 × 10^−01^	6.3626 × 10^−01^	6.4090 × 10^−01^	1.7552 × 10^−03^	6.4001 × 10^−03^
1.5135 × 10^+01^	5.3450 × 10^−01^	5.3603 × 10^−01^	5.3851 × 10^−01^	1.5327 × 10^−03^	4.0146 × 10^−03^
1.5531 × 10^+01^	4.2750 × 10^−01^	4.3000 × 10^−01^	4.3013 × 10^−01^	2.5043 × 10^−03^	2.6312 × 10^−03^
1.5893 × 10^+01^	3.1850 × 10^−01^	3.1984 × 10^−01^	3.1773 × 10^−01^	1.3443 × 10^−03^	−7.6853 × 10^−04^
1.6223 × 10^+01^	2.0850 × 10^−01^	2.0881 × 10^−01^	2.0516 × 10^−01^	3.1405 × 10^−04^	−3.3390 × 10^−03^
1.6524 × 10^+01^	1.0100 × 10^−01^	9.7726 × 10^−02^	9.3264 × 10^−02^	−3.2735 × 10^−03^	−7.7359 × 10^−03^
1.6799 × 10^+01^	−0.0080	−7.1079 × 10^−03^	−1.0438 × 10^−02^	8.9207 × 10^−04^	−2.4375 × 10^−03^
1.7050 × 10^+01^	−0.1110	−1.1026 × 10^−01^	−1.1172 × 10^−01^	7.3550 × 10^−04^	−7.1520 × 10^−04^
1.7279 × 10^+01^	−0.2090	−2.0937 × 10^−01^	−2.0794 × 10^−01^	−3.7462 × 10^−04^	1.0641 × 10^−03^
1.7489 × 10^+01^	−0.3030	−3.0209 × 10^−01^	−2.9649 × 10^−01^	9.0695 × 10^−04^	6.5099 × 10^−03^

**Table 12 sensors-22-06989-t012:** PWP201 photovoltaic panel parameters and range.

Algorithm	I_ph_ [A]	I_o1_ [A]	n_1_	R_s_ [Ω]	R_sh_ [Ω]	I_o2_ [A]	n_2_
Range Set SDM	0–2	1 × 10^−12^– 1 × 10^−05^	1–2 × 36	0–2	0–1000		
BMOA SDM	1.0312 × 10^+00^	4.5218 × 10^−06^	4.9675 × 10^+01^	1.1683 × 10^+00^	9.9978 × 10^+02^		
Range Set DDM	0–2	1 × 10^−12^– 1 × 10^−05^	1–2 × 36	0–2	0–1000	1 × 10^−12^– 1 × 10^−05^	1–2 × 36
BMOA DDM	1.0411 × 10^+00^	1.0853 × 10^−06^	4.4566 × 10^+01^	1.3052 × 10^+00^	3.7451 × 10^+02^	1.0000 × 10^−12^	3.1753 × 10^+01^

**Table 13 sensors-22-06989-t013:** PWP201 photovoltaic panel (SDM and DDM) statistical tests.

Algorithms	RMSE	MAPE	MSE	MAE	MBE	MRE	Model
**BMOA**	**2.6399 × 10^−03^**	**3.3892 × 10^−01^**	**6.9691 × 10^−06^**	**2.1755 × 10^−03^**	**1.4890 × 10^−05^**	**3.3892 × 10^−03^**	SDM
HSDA	2.4250 × 10^−03^	5.9965 × 10^−01^	5.8809 × 10^−06^	1.9569 × 10^−03^	−9.493 × 10^−10^	5.9965 × 10^−03^	
PS	2.0496 × 10^−02^	2.3813 × 10^+01^	4.2009 × 10^−04^	1.4330 × 10^−02^	1.3837 × 10^−02^	2.3814 × 10^−01^	
GCPSO	2.63132 × 10^−03^	3.2156 × 10^−01^	6.92121 × 10^−06^	2.149 × 10^−03^	−3.67245 × 10^−05^	3.2156 × 10^−01^	
WDOWOAPSO	2.63133 × 10^−03^	3.2095 × 10^−01^	6.92121 × 10^−06^	2.1489 × 10^−03^	−3.64647 × 10^−05^	3.2095 × 10^−01^	
MPCOA	3.78232 × 10^−03^	8.11566 × 10^−01^	8.8547 × 10^−06^	3.031 × 10^−^^03^	1.76485 × 10^−03^	8.11566 × 10^−01^	
SA	2.701 × 10^−03^	5.98184 × 10^−01^	5.88229 × 10^−06^	1.9523 × 10^−03^	1.79933 × 10^−05^	5.98184 × 10^−03^	
FPA	2.7425 × 10^−03^	6.11566 × 10^−01^	6.08547 × 10^−06^	2.17225 × 10^−03^	1.81613 × 10^−05^	6.11566 × 10^−03^	
EEGWO	9.7739 × 10^−02^	2.09967 × 10^+01^	9.5529 × 10^−03^	8.49683 × 10^−02^	6.71825 × 10^−02^	2.09967 × 10^−01^	
5P	4.1251 × 10^−03^	1.97806	4.04863 × 10^−06^	3.33409 × 10^−03^	1.80911 × 10^−03^	1.97806 × 10^−02^	
**BMOA**	**4.1767 × 10^−03^**	**5.6796 × 10^−01^**	**1.7445 × 10^−05^**	**3.6456 × 10^−03^**	**5.3913 × 10^−04^**	**5.6796 × 10^−03^**	DDM
HSDA	2.5877 × 10^−03^	2.8695 × 10^−01^	1.2221 × 10^−05^	2.14548 × 10^−03^	−5.3062 × 10^−05^	2.8495 × 10^−03^	
PS	2.53734 × 10^−02^	2.67472 × 10^+00^	2.3189 × 10^−04^	1.91799 × 10^−02^	6.14798 × 10^−03^	2.67472 × 10^−02^	
CPSO	7.087 × 10^−03^	9.76652 × 10^−01^	1.93012 × 10^−05^	5.11339 × 10^−03^	4.2174 × 10^−03^	9.76652 × 10^−03^	
TVACPSO	7.0917 × 10^−03^	1.01837 × 10^+00^	9.23149 × 10^−05^	5.11577 × 10^−03^	4.2201 × 10^−03^	1.01837 × 10^−02^	
SA	1.6644 × 10^−02^	4.1232 × 10^+00^	2.7801 × 10^−04^	1.00216 × 10^−02^	9.24536 × 10^−03^	4.1232 × 10^−02^	

**Table 14 sensors-22-06989-t014:** (I, V) points of Sharp ND-R250A5 photovoltaic panel.

Measured Data		SDM-BMOA	DDM-BMOA	Error Values	
V [V]	I [A]	I_c_ [A]	I_c_ [A]	SDM	DDM
0.0000 × 10^+00^	9.1500 × 10^+00^	9.2260 × 10^+00^	9.0490 × 10^+00^	7.5954 × 10^−02^	−1.0099 × 10^−01^
7.7100 × 10^+00^	9.1400 × 10^+00^	9.2228 × 10^+00^	9.0235 × 10^+00^	8.2809 × 10^−02^	−1.1648 × 10^−01^
1.0980 × 10^+01^	9.1200 × 10^+00^	9.2170 × 10^+00^	9.0124 × 10^+00^	9.7020 × 10^−02^	−1.0761 × 10^−01^
1.4550 × 10^+01^	9.1100 × 10^+00^	9.1927 × 10^+00^	8.9972 × 10^+00^	8.2675 × 10^−02^	−1.1279 × 10^−01^
1.6360 × 10^+01^	9.1000 × 10^+00^	9.1582 × 10^+00^	8.9835 × 10^+00^	5.8200 × 10^−02^	−1.1648 × 10^−01^
1.8000 × 10^+01^	9.0700 × 10^+00^	9.0957 × 10^+00^	8.9585 × 10^+00^	2.5670 × 10^−02^	−1.1145 × 10^−01^
1.9150 × 10^+01^	9.0200 × 10^+00^	9.0202 × 10^+00^	8.9243 × 10^+00^	1.6452 × 10^−04^	−9.5730 × 10^−02^
2.0040 × 10^+01^	8.9500 × 10^+00^	8.9343 × 10^+00^	8.8797 × 10^+00^	−1.5716 × 10^−02^	−7.0252 × 10^−02^
2.0870 × 10^+01^	8.8600 × 10^+00^	8.8239 × 10^+00^	8.8149 × 10^+00^	−3.6054 × 10^−02^	−4.5124 × 10^−02^
2.1670 × 10^+01^	8.7300 × 10^+00^	8.6831 × 10^+00^	8.7217 × 10^+00^	−4.6926 × 10^−02^	−8.3171 × 10^−03^
2.2360 × 10^+01^	8.5800 × 10^+00^	8.5281 × 10^+00^	8.6080 × 10^+00^	−5.1924 × 10^−02^	2.7988 × 10^−02^
2.3020 × 10^+01^	8.4000 × 10^+00^	8.3455 × 10^+00^	8.4617 × 10^+00^	−5.4537 × 10^−02^	6.1684 × 10^−02^
2.3620 × 10^+01^	8.2000 × 10^+00^	8.1468 × 10^+00^	8.2902 × 10^+00^	−5.3186 × 10^−02^	9.0210 × 10^−02^
2.4150 × 10^+01^	8.0000 × 10^+00^	7.9409 × 10^+00^	8.1008 × 10^+00^	−5.9121 × 10^−02^	1.0077 × 10^−01^
2.4610 × 10^+01^	7.8000 × 10^+00^	7.7392 × 10^+00^	7.9063 × 10^+00^	−6.0845 × 10^−02^	1.0630 × 10^−01^
2.5020 × 10^+01^	7.6000 × 10^+00^	7.5409 × 10^+00^	7.7082 × 10^+00^	−5.9135 × 10^−02^	1.0823 × 10^−01^
2.5390 × 10^+01^	7.4000 × 10^+00^	7.3474 × 10^+00^	7.5099 × 10^+00^	−5.2611 × 10^−02^	1.0991 × 10^−01^
2.5750 × 10^+01^	7.2000 × 10^+00^	7.1403 × 10^+00^	7.2910 × 10^+00^	−5.9705 × 10^−02^	9.0985 × 10^−02^
2.6380 × 10^+01^	6.8000 × 10^+00^	6.7489 × 10^+00^	6.8694 × 10^+00^	−5.1104 × 10^−02^	6.9380 × 10^−02^
2.6940 × 10^+01^	6.4000 × 10^+00^	6.3679 × 10^+00^	6.4526 × 10^+00^	−3.2113 × 10^−02^	5.2583 × 10^−02^
2.7460 × 10^+01^	6.0000 × 10^+00^	5.9823 × 10^+00^	6.0263 × 10^+00^	−1.7691 × 10^−02^	2.6274 × 10^−02^
2.7940 × 10^+01^	5.6000 × 10^+00^	5.6051 × 10^+00^	5.6111 × 10^+00^	5.0724 × 10^−03^	1.1129 × 10^−02^
2.8400 × 10^+01^	5.2000 × 10^+00^	5.2172 × 10^+00^	5.1841 × 10^+00^	1.7226 × 10^−02^	−1.5873 × 10^−02^
2.8840 × 10^+01^	4.8000 × 10^+00^	4.8244 × 10^+00^	4.7548 × 10^+00^	2.4374 × 10^−02^	−4.5221 × 10^−02^
2.9250 × 10^+01^	4.4000 × 10^+00^	4.4526 × 10^+00^	4.3621 × 10^+00^	5.2620 × 10^−02^	−3.7912 × 10^−02^
2.9660 × 10^+01^	4.0000 × 10^+00^	4.0494 × 10^+00^	3.9328 × 10^+00^	4.9441 × 10^−02^	−6.7156 × 10^−02^
3.0050 × 10^+01^	3.6000 × 10^+00^	3.6585 × 10^+00^	3.5306 × 10^+00^	5.8460 × 10^−02^	−6.9446 × 10^−02^
3.0440 × 10^+01^	3.2000 × 10^+00^	3.2379 × 10^+00^	3.0962 × 10^+00^	3.7926 × 10^−02^	−1.0385 × 10^−01^
3.0810 × 10^+01^	2.8000 × 10^+00^	2.8387 × 10^+00^	2.7041 × 10^+00^	3.8704 × 10^−02^	−9.5852 × 10^−02^
3.1170 × 10^+01^	2.4000 × 10^+00^	2.4410 × 10^+00^	2.3267 × 10^+00^	4.0994 × 10^−02^	−7.3275 × 10^−02^
3.1520 × 10^+01^	2.0000 × 10^+00^	2.0483 × 10^+00^	1.9695 × 10^+00^	4.8280 × 10^−02^	−3.0503 × 10^−02^
3.1880 × 10^+01^	1.6000 × 10^+00^	1.6013 × 10^+00^	1.5478 × 10^+00^	1.3221 × 10^−03^	−5.2168 × 10^−02^
3.2220 × 10^+01^	1.2000 × 10^+00^	1.1933 × 10^+00^	1.1964 × 10^+00^	−6.6965 × 10^−03^	−3.6094 × 10^−03^
3.2550 × 10^+01^	8.0000 × 10^−01^	7.9840 × 10^−01^	8.7793 × 10^−01^	−1.5976 × 10^−03^	7.7926 × 10^−02^
3.2890 × 10^+01^	4.0000 × 10^−01^	3.4739 × 10^−01^	4.9477 × 10^−01^	−5.2606 × 10^−02^	9.4767 × 10^−02^
3.3220 × 10^+01^	0.0000 × 10^+00^	−8.9116 × 10^−02^	1.4756 × 10^−01^	−8.9116 × 10^−02^	1.4756 × 10^−01^

**Table 15 sensors-22-06989-t015:** Sharp ND-R250A5 photovoltaic panel parameters and range.

Algorithm	I_ph_ [A]	I_o1_ [A]	n_1_	R_s_ [Ω]	R_sh_ [Ω]	I_o2_ [A]	n_2_
Range Set SDM	0–10	1 × 10^−12^– 1 × 10^−05^	1–2 × 60	0–1	0–5500		
BMOA SDM	9.2269 × 10^+00^	9.8302 × 10^−06^	8.4341 × 10^+01^	5.3520 × 10^−01^	5.5000 × 10^+03^		
Range Set DDM	0–10	1 × 10^−12^– 1 × 10^−05^	1–2 × 36	0–1	0–1000	1 × 10^−12^– 1 × 10^−05^	1–2 × 36
BMOA DDM	9.0690 × 10^+00^	1.0000 × 10^−12^	4.9966 × 10^+01^	6.6079 × 10^−01^	3.0291 × 10^+02^	1.3240 × 10^−08^	5.7132 × 10^+01^

**Table 16 sensors-22-06989-t016:** Sharp ND-R250A5 photovoltaic panel (SDM and DDM) statistical tests.

Algorithms	RMSE	MAPE	MSE	MAE	MBE	MRE	Model
**BMOA**	**5.0937 × 10^−02^**	**7.4435 × 10^−01^**	**2.5946 × 10^−03^**	**4.4378 × 10^−02^**	**−1.0478 × 10^−04^**	**7.4435 × 10^−03^**	SDM
HSDA	1.1403 × 10^−02^	6.6463 × 10^+02^	1.3005 × 10^−04^	9.4065 × 10^−03^	8.1694 × 10^−07^	6.6464 × 10^+00^	
GCPSO	1.1532 × 10^−02^	4.1184 × 10^+02^	2.92161 × 10^−04^	9.55131 × 10^−03^	2.93882 × 10^−06^	4.1184 × 10^+00^	
5P	8.34268 × 10^−02^	8.14341 × 10^−01^	3.87231 × 10^−03^	6.0672 × 10^−02^	5.3637 × 10^−02^	8.14341 × 10^−03^	
**BMOA**	**8.2295 × 10^−02^**	**1.2374 × 10^+00^**	**6.7725 × 10^−03^**	**7.3772 × 10^−02^**	**−8.4553 × 10^−03^**	**1.2374 × 10^−02^**	DDM
HSDA	1.1222 × 10^−02^	4.3452 × 10^−01^	1.2012 × 10^−04^	9.4042 × 10^−03^	1.17243 × 10^−06^	4.3452 × 10^−03^	
GCPSO	1.6644 × 10^−02^	4.1232 × 10^+00^	2.6401 × 10^−04^	9.55131 × 10^−03^	3.13956 × 10^−06^	4.1232 × 10^−02^	

**Table 17 sensors-22-06989-t017:** (I, V) points of Kyocera KC200GT photovoltaic panel.

Measured Data		SDM-BMOA	DDM-BMOA	Error Values	
V [V]	I [A]	I_c_ [A]	I_c_ [A]	SDM	DDM
4.0900 × 10^−02^	8.1761 × 10^+00^	8.1391 × 10^+00^	8.1353 × 10^+00^	−3.7030 × 10^−02^	−4.0827 × 10^−02^
1.0472 × 10^+00^	8.1761 × 10^+00^	8.1371 × 10^+00^	8.1330 × 10^+00^	−3.9047 × 10^−02^	−4.3096 × 10^−02^
2.0534 × 10^+00^	8.1761 × 10^+00^	8.1350 × 10^+00^	8.1307 × 10^+00^	−4.1067 × 10^−02^	−4.5368 × 10^−02^
3.0707 × 10^+00^	8.1364 × 10^+00^	8.1330 × 10^+00^	8.1284 × 10^+00^	−3.4093 × 10^−03^	−7.9659 × 10^−03^
4.0769 × 10^+00^	8.1364 × 10^+00^	8.1310 × 10^+00^	8.1262 × 10^+00^	−5.4390 × 10^−03^	−1.0248 × 10^−02^
4.9082 × 10^+00^	8.1364 × 10^+00^	8.1293 × 10^+00^	8.1243 × 10^+00^	−7.1226 × 10^−03^	−1.2141 × 10^−02^
6.0129 × 10^+00^	8.1178 × 10^+00^	8.1270 × 10^+00^	8.1217 × 10^+00^	9.2278 × 10^−03^	3.9313 × 10^−03^
7.0192 × 10^+00^	8.1178 × 10^+00^	8.1250 × 10^+00^	8.1194 × 10^+00^	7.1555 × 10^−03^	1.6052 × 10^−03^
8.0255 × 10^+00^	8.1178 × 10^+00^	8.1229 × 10^+00^	8.1170 × 10^+00^	5.0520 × 10^−03^	−7.5308 × 10^−04^
9.0427 × 10^+00^	8.0781 × 10^+00^	8.1207 × 10^+00^	8.1146 × 10^+00^	4.2581 × 10^−02^	3.6518 × 10^−02^
1.0049 × 10^+01^	8.0781 × 10^+00^	8.1185 × 10^+00^	8.1121 × 10^+00^	4.0356 × 10^−02^	3.4035 × 10^−02^
1.1066 × 10^+01^	8.0781 × 10^+00^	8.1161 × 10^+00^	8.1095 × 10^+00^	3.7993 × 10^−02^	3.1411 × 10^−02^
1.1974 × 10^+01^	8.0781 × 10^+00^	8.1138 × 10^+00^	8.1070 × 10^+00^	3.5738 × 10^−02^	2.8921 × 10^−02^
1.2991 × 10^+01^	8.0595 × 10^+00^	8.1111 × 10^+00^	8.1040 × 10^+00^	5.1569 × 10^−02^	4.4485 × 10^−02^
1.3998 × 10^+01^	8.0595 × 10^+00^	8.1079 × 10^+00^	8.1006 × 10^+00^	4.8441 × 10^−02^	4.1090 × 10^−02^
1.5015 × 10^+01^	8.0385 × 10^+00^	8.1042 × 10^+00^	8.0966 × 10^+00^	6.5685 × 10^−02^	5.8061 × 10^−02^
1.6021 × 10^+01^	8.0385 × 10^+00^	8.0996 × 10^+00^	8.0917 × 10^+00^	6.1064 × 10^−02^	5.3166 × 10^−02^
1.7027 × 10^+01^	8.0199 × 10^+00^	8.0936 × 10^+00^	8.0854 × 10^+00^	7.3679 × 10^−02^	6.5503 × 10^−02^
1.8045 × 10^+01^	8.0199 × 10^+00^	8.0854 × 10^+00^	8.0769 × 10^+00^	6.5492 × 10^−02^	5.7038 × 10^−02^
1.9051 × 10^+01^	8.0012 × 10^+00^	8.0741 × 10^+00^	8.0654 × 10^+00^	7.2887 × 10^−02^	6.4162 × 10^−02^
2.0068 × 10^+01^	7.9802 × 10^+00^	8.0577 × 10^+00^	8.0487 × 10^+00^	7.7491 × 10^−02^	6.8511 × 10^−02^
2.1074 × 10^+01^	7.9802 × 10^+00^	8.0339 × 10^+00^	8.0247 × 10^+00^	5.3731 × 10^−02^	4.4546 × 10^−02^
2.2081 × 10^+01^	7.9616 × 10^+00^	7.9988 × 10^+00^	7.9895 × 10^+00^	3.7173 × 10^−02^	2.7865 × 10^−02^
2.2999 × 10^+01^	7.9219 × 10^+00^	7.9517 × 10^+00^	7.9424 × 10^+00^	2.9839 × 10^−02^	2.0545 × 10^−02^
2.4017 × 10^+01^	7.9219 × 10^+00^	7.8743 × 10^+00^	7.8653 × 10^+00^	−4.7571 × 10^−02^	−5.6563 × 10^−02^
2.5023 × 10^+01^	7.8823 × 10^+00^	7.7586 × 10^+00^	7.7503 × 10^+00^	−1.2372 × 10^−01^	−1.3196 × 10^−01^
2.6029 × 10^+01^	7.7844 × 10^+00^	7.5838 × 10^+00^	7.5770 × 10^+00^	−2.0063 × 10^−01^	−2.0743 × 10^−01^
2.7046 × 10^+01^	7.5699 × 10^+00^	7.3169 × 10^+00^	7.3126 × 10^+00^	−2.5296 × 10^−01^	−2.5730 × 10^−01^
2.8053 × 10^+01^	7.1385 × 10^+00^	6.9214 × 10^+00^	6.9205 × 10^+00^	−2.1707 × 10^−01^	−2.1797 × 10^−01^
2.9070 × 10^+01^	6.4717 × 10^+00^	6.3249 × 10^+00^	6.3283 × 10^+00^	−1.4684 × 10^−01^	−1.4341 × 10^−01^
3.0076 × 10^+01^	5.4155 × 10^+00^	5.4574 × 10^+00^	5.4639 × 10^+00^	4.1890 × 10−02	4.8380 × 10^−02^
3.1181 × 10^+01^	3.8091 × 10^+00^	4.0444 × 10^+00^	4.0483 × 10^+00^	2.3528 × 10−01	2.3921 × 10^−01^
3.2100 × 10^+01^	2.1443 × 10^+00^	2.3452 × 10^+00^	2.3357 × 10^+00^	2.0092 × 10^−01^	1.9144 × 10^−01^
3.2920 × 10^+01^	4.4000 × 10^−01^	2.6808 × 10^−01^	2.3004 × 10^−01^	−1.7192 × 10^−01^	−2.0996 × 10^−01^

**Table 18 sensors-22-06989-t018:** Kyocera KC200GT photovoltaic parameters and range.

Algorithm	I_ph_ [A]	I_o1_ [A]	n_1_	R_s_ [Ω]	R_sh_ [Ω]	I_o2_ [A]	n_2_
Range Set SDM	0–10	1 × 10^−12^– 1 × 10^−05^	1–128	0–0.5	0–500		
BMOA SDM	8.1400 × 10^+00^	7.6704 × 10^−06^	9.2687 × 10^+01^	4.9389 × 10^−02^	5.0000 × 10^+02^		
Range Set DDM	0–10	1 × 10^−12^– 1 × 10^−05^	1–128	0–0.5	0–500	1 × 10^−12^– 1 × 10^−05^	1–128
BMOA DDM	8.1362 × 10^+00^	1.4810 × 10^−06^	9.8760 × 10^+01^	4.2709 × 10^−02^	4.4436 × 10^+02^	6.8870 × 10^−06^	9.2509 × 10^+01^

**Table 19 sensors-22-06989-t019:** Kyocera KC200GT photovoltaic panel (SDM and DDM) statistical tests.

Algorithms	RMSE	MAPE	MSE	MAE	MBE	MRE	Model
**BMOA**	**1.0400 × 10^−01^**	**1.0338 × 10^+00^**	**1.0817 × 10^−02^**	**7.6091 × 10^−02^**	**−1.7187 × 10^−05^**	**1.0338 × 10^−02^**	SDM
HSDA	1.8439 × 10^−02^	3.2289 × 10^−01^	3.4001 × 10^−04^	1.4275 × 10^−02^	1.70154 × 10^−08^	3.2289 × 10^−03^	
ICA	3.2255 × 10^−01^	1.0709 × 10^+01^	1.0404 × 10^−01^	1.8333 × 10^−01^	9.76795 × 10^−02^	1.0709 × 10^−01^	
WDO	1.6121 × 10^−01^	2.3813 × 10^+01^	4.2009 × 10^−02^	1.00538 × 10^−01^	−3.5713 × 10^−02^	2.3814 × 10^−01^	
**BMOA**	**1.0542 × 10^−01^**	**1.0171 × 10^+00^**	**1.1113 × 10^−02^**	**7.4865 × 10^−02^**	**−6.6048 × 10^−03^**	**1.0171 × 10^−02^**	DDM
HSDA	1.1903 × 10^−02^	2.7665 × 10^−01^	2.9721 × 10^−04^	9.09918 × 10^−03^	−1.0616 × 10^−04^	2.7665 × 10^−03^	
ICA	3.0128 × 10^−01^	1.0021 × 10^+01^	0.9812 × 10^−01^	1.6543 × 10^−01^	8.91235 × 10^−02^	1.0021 × 10^−01^	

**Table 20 sensors-22-06989-t020:** Computing time-based analysis with two computers with distinct features.

Computer	Models	Computing Time (s)					
		Monocrystalline	aSi	RTC	SHARP ND-R250A5	PWP201	KYOCERA KC200GT
**HP Laptop**	**SDM**	26.058207	25.83922	16.14553	24.533244	30.897654	36.944390
	**DDM**	22.313337	19.70922	24.96470	20.947452	35.317376	44.448694
**Asus Computer**	**SDM**	5.578089	4.746904	5.414492	6.341659	4.671119	5.169759
	**DDM**	5.417908	5.520116	5.601866	6.048046	7.591440	7.449753

**Table 21 sensors-22-06989-t021:** Number of the iteration for the algorithms.

Algorithms	BMOA	EEGWO	CPSO	CSO	FPA	MPCOA	ICA	GWO	WOA	WDO	WDOWOAPSO	CS	PSO
No. of Iterations	1000	50,000	45,000	15,000	25,000	250,000	100	50,000	50,000	15,000	15,000	15,000	15,000

**Table 22 sensors-22-06989-t022:** Influence of the epochs and population on the statistical tests, monocrystalline silicon photovoltaic cell.

Epochs/Population	RMSE	MAPE	MSE	MAE	MBE	MRE	Time (s)	Model
1000/250	1.8456 × 10^−03^	5.1804 × 10^−01^	3.4062 × 10^−06^	1.5580 × 10^−03^	5.1804 × 10^−01^	5.1804 × 10^−03^	5.578089	SDM
500/250	1.9272 × 10^−03^	5.3858 × 10^−01^	3.7139 × 10^−06^	1.6198 × 10^−03^	4.9017 × 10^−05^	5.3858 × 10^−03^	3.199777	
1000/30	6.4535 × 10^−04^	1.7221 × 10^−01^	4.1647 × 10^−07^	5.1793 × 10^−04^	−8.6553 × 10^−07^	1.7221 × 10^−03^	1.954307	
1000/15	9.2551 × 10^−02^	2.5969 × 10^+01^	8.5656 × 10^−03^	7.8104 × 10^−02^	−7.9338 × 10^−03^	2.5969 × 10_−01_	1.827962	
1000/250	1.3754 × 10^−03^	4.0682 × 10^−01^	1.8918 × 10^−06^	1.2235 × 10^−03^	2.4771 × 10^−04^	4.0682 × 10^−03^	5.417908	DDM
500/250	1.5272 × 10^−03^	4.5134 × 10^−01^	2.3325 × 10^−06^	1.3574 × 10^−03^	−2.5770 × 10^−04^	4.5134 × 10^−03^	3.616986	
1000/30	1.4729 × 10^−02^	4.1186 × 10^+00^	2.1695 × 10^−04^	1.2387 × 10^−02^	1.3994 × 10^−04^	4.1186 × 10^−02^	2.028410	
1000/15	2.5023 × 10^−03^	6.9126 × 10^−01^	6.2616 × 10^−06^	2.0790 × 10^−03^	1.9232 × 10^−05^	6.9126 × 10^−03^	1.940299	

**Table 23 sensors-22-06989-t023:** Influence of the epochs and population on the statistical tests, amorphous silicon photovoltaic cell.

Epochs/Population	RMSE	MAPE	MSE	MAE	MBE	MRE	Time (s)	Model
1000/250	8.2115 × 10^−05^	7.3431 × 10^−01^	6.7430 × 10^−09^	6.9524 × 10^−05^	−1.6201 × 10^−06^	7.3431 × 10^−03^	4.746904	SDM
500/250	8.6880 × 10^−05^	6.5991 × 10^−01^	7.5482 × 10^−09^	6.2480 × 10^−05^	4.5950 × 10^−06^	6.5991 × 10^−03^	3.364575	
1000/30	6.9075 × 10^−05^	5.6519 × 10^−01^	4.7713 × 10^−09^	5.3512 × 10^−05^	3.4706 × 10^−07^	5.6519 × 10^−03^	2.461296	
1000/15	4.6596 × 10^−05^	4.2561 × 10^−01^	2.1712 × 10^−09^	4.0297 × 10^−05^	1.4653 × 10^−08^	4.2561 × 10^−03^	1.934502	
1000/250	5.3511 × 10^−05^	4.6531 × 10^−01^	2.8634 × 10^−09^	4.4055 × 10^−05^	3.7606 × 10^−06^	4.6531 × 10^−03^	5.520116	DDM
500/250	5.5825 × 10^−05^	4.7933 × 10^−01^	3.1165 × 10^−09^	4.5382 × 10^−05^	3.1733 × 10^−06^	4.7933 × 10^−03^	3.383654	
1000/30	6.8260 × 10^−05^	6.0831 × 10^−01^	4.6595 × 10^−09^	5.7595 × 10^−05^	−2.2826 × 10^−06^	6.0831 × 10^−03^	2.074660	
1000/15	4.7475 × 10^−05^	4.1859 × 10^−01^	2.2538 × 10^−09^	3.9632 × 10^−05^	−1.9501 × 10^−07^	4.1859 × 10^−03^	1.795082	

**Table 24 sensors-22-06989-t024:** Influence of the epochs and population on the statistical tests—RTC silicon photovoltaic cell.

Epochs/Population	RMSE	MAPE	MSE	MAE	MBE	MRE	Time (s)	Model
1000/250	7.0675 × 10^−03^	1.0939 × 10^+00^	4.9949 × 10^−05^	6.0335 × 10^−03^	1.0712 × 10^−04^	1.0939 × 10^−02^	5.414492	SDM
500/250	7.1319 × 10^−03^	1.0895 × 10^+00^	5.0865 × 10^−05^	6.0093 × 10^−03^	−1.6424 × 10^−05^	1.0895 × 10^−02^	3.072668	
1000/30	3.3436 × 10^−03^	5.0710 × 10^−01^	1.1180 × 10^−05^	2.7970 × 10^−03^	−6.0565 × 10^−06^	5.0710 × 10^−03^	1.938164	
1000/15	2.1871 × 10^−02^	3.0671 × 10^+00^	4.7834 × 10^−04^	1.6917 × 10^−02^	−3.2835 × 10^−10^	3.0671 × 10^−02^	2.019497	
1000/250	5.6269 × 10^−03^	8.3504 × 10^−01^	3.1662 × 10^−05^	4.6059 × 10^−03^	3.8898 × 10^−04^	8.3504 × 10^−03^	5.601866	DDM
500/250	5.7223 × 10^−03^	8.7305 × 10^−01^	3.2744 × 10^−05^	4.8155 × 10^−03^	−2.2145 × 10^−04^	8.7305 × 10^−03^	3.552039	
1000/30	3.9968 × 10^−03^	6.1808 × 10^−01^	1.5975 × 10^−05^	3.4092 × 10^−03^	1.0824 × 10^−04^	6.1808 × 10^−03^	2.018845	
1000/15	1.8145 × 10^−02^	2.2538 × 10^+00^	3.2925 × 10^−04^	1.2431 × 10^−02^	−2.8779 × 10^−10^	2.2538 × 10^−02^	1.910127	

**Table 25 sensors-22-06989-t025:** Influence of the epochs and population on the statistical tests, PWP201 photovoltaic panel.

Epochs/Population	RMSE	MAPE	MSE	MAE	MBE	MRE	Time (s)	Model
1000/250	2.6399 × 10^−03^	3.3892 × 10^−01^	6.9691 × 10^−06^	2.1755 × 10^−03^	1.4890 × 10^−05^	3.3892 × 10^−03^	4.671119	SDM
500/250	2.9840 × 10^−03^	4.0219 × 10^−01^	8.9043 × 10^−06^	2.5816 × 10^−03^	4.4192 × 10^−04^	4.0219 × 10^−03^	4.458341	
1000/30	4.2114 × 10^−03^	5.4951 × 10^−01^	1.7736 × 10^−05^	3.5272 × 10^−03^	−1.2358 × 10^−05^	5.4951 × 10^−03^	1.986885	
1000/15	4.1511 × 10^−03^	5.4081 × 10^−01^	1.7232 × 10^−05^	3.4713 × 10^−03^	−2.1813 × 10^−05^	5.4081 × 10^−03^	1.722779	
1000/250	4.1767 × 10^−03^	5.6796 × 10^−01^	1.7445 × 10^−05^	3.6456 × 10^−03^	5.3913 × 10^−04^	5.6796 × 10^−03^	7.591440	DDM
500/250	4.9159 × 10^−03^	6.4916 × 10^−01^	2.4166 × 10^−05^	4.1669 × 10^−03^	6.8058 × 10^−04^	6.4916 × 10^−03^	6.074092	
1000/30	2.7894 × 10^−03^	3.5115 × 10^−01^	7.7808 × 10^−06^	2.2540 × 10^−03^	2.3122 × 10^−06^	3.5115 × 10^−03^	4.181851	
1000/15	3.5714 × 10^−03^	4.5709 × 10^−01^	1.2755 × 10^−05^	2.9340 × 10^−03^	−2.4876 × 10^−06^	4.5709 × 10^−03^	3.858213	

**Table 26 sensors-22-06989-t026:** Influence of the epochs and population on the statistical tests, Sharp ND-R250A5 photovoltaic panel.

Epochs/Population	RMSE	MAPE	MSE	MAE	MBE	MRE	Time (s)	Model
1000/250	5.0937 × 10^−02^	7.4435 × 10^−01^	2.5946 × 10^−03^	4.4378 × 10^−02^	−1.0478 × 10^−04^	7.4435 × 10^−03^	6.341659	SDM
500/250	5.1162 × 10^−02^	7.5606 × 10^−01^	2.6176 × 10^−03^	4.5076 × 10^−02^	−8.4940 × 10^−04^	7.5606 × 10^−03^	3.787169	
1000/30	4.9525 × 10^−02^	7.2640 × 10^−01^	2.4527 × 10^−03^	4.3307 × 10^−02^	−2.9957 × 10^−06^	7.2640 × 10^−03^	2.256192	
1000/15	2.9627 × 10^−02^	4.3841 × 10^−01^	8.7774 × 10^−04^	2.6138 × 10^−02^	1.1430 × 10^−05^	4.3841 × 10^−03^	2.168015	
1000/250	8.2295 × 10^−02^	1.2374 × 10^+00^	6.7725 × 10^−03^	7.3772 × 10^−02^	−8.4553 × 10^−03^	1.2374 × 10^−02^	6.048046	DDM
500/250	8.7270 × 10^−02^	1.3148 × 10^+00^	7.6160 × 10^−03^	7.8389 × 10^−02^	−1.8889 × 10^−02^	1.3148 × 10^−02^	3.774458	
1000/30	1.7599 × 10^−01^	2.6472 × 10^+00^	3.0974 × 10^−02^	1.5782 × 10^−01^	−3.9619 × 10^−03^	2.6472 × 10^−02^	2.300886	
1000/15	1.2472 × 10^−02^	1.7318 × 10^−01^	1.5556 × 10^−04^	1.0325 × 10^−02^	−1.0157 × 10^−04^	1.7318 × 10^−03^	1.993856	

**Table 27 sensors-22-06989-t027:** Influence of the epochs and population on the statistical tests, Kyocera KC200GT photovoltaic panel.

Epochs/Population	RMSE	MAPE	MSE	MAE	MBE	MRE	Time (s)	Model
1000/250	1.0400 × 10^−01^	1.0338 × 10^+00^	1.0817 × 10^−02^	7.6091 × 10^−02^	−1.7187 × 10^−05^	1.0338 × 10^−02^	5.169759	SDM
500/250	1.04323 × 10^−01^	1.05583938	1.09 × 10^−02^	0.07771654	5.86 × 10^−03^	0.01055839	2.544905	
1000/30	1.0713 × 10^−01^	1.0597 × 10^+00^	1.1478 × 10^−02^	7.8002 × 10^−02^	−2.8056 × 10^−09^	1.0597 × 10^−02^	1.931199	
1000/15	1.0647 × 10^−01^	1.0553 × 10^+00^	1.1335 × 10^−02^	7.7675 × 10^−02^	5.8815 × 10^−04^	1.0553 × 10^−02^	1.915780	
1000/250	1.0542 × 10^−01^	1.0171 × 10^+00^	1.1113 × 10^−02^	7.4865 × 10^−02^	−6.6048 × 10^−03^	1.0171 × 10^−02^	7.449753	DDM
500/250	1.07877 × 10^−01^	1.06580371	1.16 × 10^−02^	0.07844998	1.03 × 10^−02^	0.01065803	5.249409	
1000/30	1.0713 × 10^−01^	1.0598 × 10^+00^	1.1478 × 10^−02^	7.8007 × 10^−02^	1.9315 × 10^−05^	1.0598 × 10^−02^	4.566931	
1000/15	1.0389 × 10^−01^	1.0340 × 10^+00^	1.0792 × 10^−02^	7.6109 × 10^−02^	1.1422 × 10^−04^	1.0340 × 10^−02^	3.901237	

## Data Availability

Not applicable.

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
