# Peer review of "Barnacles Mating Optimizer Algorithm to Extract the Parameters of the Photovoltaic Cells and Panels"

_sensors, 2022, doi:10.3390/s22186989_

Round 1
Reviewer 1 Report
The topic of the paper is highly interesting, but I have serious doubts about the scientific contributions of this work. The proposed procedure consists in applying the Barnacles mating optimizer algorithm to extract PV module parameters. In my opinion, this procedure cannot be considered as a novel contribution to the state-of-the-art of the analysis of PV module.
It is suggested to use a bibliographic manager such as Mendeley and check that the metadata are complete such as: Title, authors, keywords, DOI, URL, year, and in case of journals, volume and issue
All the references cited in the text should be analysed and discussed. Lumping references like [24-26], etc., should be avoided, and instead the main contribution of each referenced paper should be summarized and discussed in a separate sentence
First of all, it is not clear what is the real advantage of the optimizer algorithm, since it builds up on previous references and it is not clear to me what is new.
The Abstract in its current from is an alternative Introduction, it should clearly describe the scope with more focusing on the proposed approach and the specific results of the study to be applied to other cases.
Introduction section should provide a critical analysis of the available and appropriate literature to identify an objective whose accomplishment will provide a significant contribution to the field. The research gap in the literature should be clearly exposed.
Author Response
Many thanks for your guidance, and for all the comments and suggestions from the reviewer of this paper. According to the reviewer’s comments and suggestions, we have carefully revised the manuscript. The revised parts have been marked on the manuscript in red.

Reviewer 2 Report
The authors of this article present analyzes related to the use of Barnacles mating optimizer algorithm to extract the parameters of the selected photovoltaic cells. The applied approach is aimed at increasing the efficiency and effectiveness in solving the optimization problems presented in the article. However, there are some considerations and questions that should be addressed and clarified:
• The literature review could be further extended and deepened.
• Where in the manuscript are the abbreviations "SDA" and "PSDA" explained?
• In my opinion, the manuscript lacks a list of the values of all control parameters for other optimization algorithms used in the calculations (eg control parameters for the algorithm: GA, SDA, 5P, PSDA, CS, SA, ICA, etc.). In addition, the population size, the maximum number of iterations should be the same for each algorithm, then the comparison of the results is more reliable. Only the number of iterations is given in table 21, and it is different for different algorithms. Other parameters are also needed.
• Have calculations been performed for a smaller number of populations (eg, less than 50 or less than 20)? Maybe the use of other heuristic algorithms would reduce the number of populations and further shorten the computation time?
• Did the Authors perform calculations for a smaller number of epochs, eg 500? How effective is the algorithm then?
• The authors write that "The novelty and contributions of this paper are: Developing and implementing a new metaheuristic algorithm for the first time to extract the five or seven parameters of three photovoltaic cells and for one photovoltaic panel - Barnacles Mating Optimizer Algorithm (BMOA) ". What's new The authors have developed / modified the BMO algorithm in comparison to the algorithm described in the cited paper [32].
• In my opinion the font of the text in the diagrams is too small, e.g. Fig. 2, Fig. 3, Fig. 4, Fig. 5, Fig. 6, Fig. 7.
• The single letters "a" or the articles "the" appear at the end of some lines. They should appear at the beginning of the next line.
Author Response

(The authors gave the same response as above.)

Round 2
Reviewer 1 Report
I have no further comments. The paper can be published as it is.
Reviewer 2 Report
Thanks for the responses. I have no more comments. Good luck